# A Review of Chipless Remote Sensing Solutions Based on RFID Technology

**DOI:** 10.3390/s19224829

**Published:** 2019-11-06

**Authors:** Kevin Mc Gee, Prince Anandarajah, David Collins

**Affiliations:** 1School of Biotechnology, Dublin City University, Dublin 9, Ireland; david.collins@dcu.ie; 2Photonics Systems and Sensing Laboratory, School of Electronic Engineering, Dublin City University, Dublin 9, Ireland; prince.anandarajah@dcu.ie

**Keywords:** chipless RFID, RFID sensors, sensing materials

## Abstract

Chipless Radio Frequency Identification (RFID) has been used in a variety of remote sensing applications and is currently a hot research topic. To date, there have been a large number of chipless RFID tags developed in both academia and in industry that boast a large variation in design characteristics. This review paper sets out to discuss the various design aspects needed in a chipless RFID sensor. Such aspects include: (1) Addressing strategies to allow for unique identification of the tag, (2) Sensing mechanisms used to allow for impedance-based response signal modulation and (3) Sensing materials to introduce the desired impedance change when under the influence of the target stimulus. From the tabular comparison of the various sensing and addressing techniques, it is concluded that although many sensors provide adequate performance characteristics, more work is needed to ensure that this technology is capable/robust enough to operate in many of the applications it has been earmarked for.

## 1. Introduction

Chipless Radio Frequency Identification (RFID) sensing is becoming a hot research topic and has attracted much attention over the past number of years [1,2,3,4]. This technology has spawned from earlier work into RFID tag design which is now a well-established field of research.

Integrating sensing elements into existing RFID technology has brought forward great possibilities in the remote sensing of temperature [5,6], humidity [7,8], strain [9,10] and other environmental stimuli. Such integrations usually involve the addition of a stimulus-sensitive coating to an existing chipless RFID tag, which results in a tag whose interrogation response is related to the stimulus of interest. Other approaches make use of stimulus-sensitive materials as the main tag material. This review paper has been compiled to assess the current state-of-the-art in this research area.

### 1.1. Overview and Organisation of Paper

This paper sets out to review the current state-of-the-art in passive RFID sensing. Although many such sensors have been found in the literature, this review will focus on the underlying designs that many of these sensors utilise, as opposed to focusing on any one specific approach.

From the literature, it would appear that many of the sensing techniques could be applied to any and indeed all of the underlying RFID tag designs and thus an emphasis will be placed on the emerging sensing techniques and less emphasis on specific RFID sensor implementations.

The paper is laid out in two main sections, Addressing and Sensing. Focus is placed on the various addressing schemes in the first section because many of the existing RFID sensors utilise an addressing scheme that is stimulus-sensitive. Within those sections, time- and spectral-based implementations are reviewed separately as their characteristics and performances vary quite significantly. Although there are fewer sensor implementations than basic RFID addressed tags, virtually any of the addressing schemes can be used along with a stimulus-sensitive material to achieve the sensing of a specific stimulus.

At this point, the authors would like to draw the attention of the reader to a recent review paper on the topic of chipless RFID addressing by Herrojo et al. [11]. This review covers a large amount of chipless RFID addressing schemes and therefore, the focus of the addressing section in this paper will be to complement the material reviewed in Reference [11]. Although several other review papers have already been published in the area of chipless RFID tagging and sensing, the current reviews are somewhat incomplete. Many reviews discuss one branch of this technology space in isolation, e.g., Surface Acoustic Wave (SAW) solutions, as seen in Reference [12] by Plessky and Reindl, or Time Domain solutions only, as seen in the work of Forouzandeh and Karmakar [13]. Similarly, other reviews focus on several branches of this domain but leave others out. Noteworthy examples of focused reviews of this sort include the work of Hashemi et al. [14] and that of Hagelauer et al. [3]. With regard to the variety of tag designs that are present in the current literature, many tags share common underlying designs, e.g., Split Ring Resonators, and thus, this review attempts to discuss all of these designs to an equal degree so that no one design dominates the content of the review paper. Therefore, this review sets out to review all of the techniques found in the RFID tag design literature and provides a tabular comparison between the various designs in each section of the review. Although no direct comparison is made between time- and spectral-based tag designs, the reader is free to draw their own conclusions on this topic, based on the content of the comparison tables in the two respective sections. One such table, in the Addressing section, references several papers that are discussed thoroughly in the main body along with other tags which are discussed in References [11,12,13]. Such tags are beyond the scope of detailed discussion in this manuscript. Although a good example of such comparison is made in Reference [11], this review paper has avoided such analysis. The reasoning behind this choice is due to the possible issues that arise when generating metrics to assess chipless RFID tags. These issues include:The measurement of encoded bit-density per unit area is difficult to quantify for various tag designs. Many tags do not include the antenna setup required to function as a remote sensor whereas others do. Therefore, a direct comparison of such tags leads to an overestimation of bit-density of tags that do not include an antenna scheme. Furthermore, alternate antenna schemes vary in size considerably, as some implementations use one antenna whereas others use two antennas. Therefore, such a metric is more of a comparison of specific tag implementations as opposed to comparison-specific RFID design methodologies.It is unclear from the current literature whether the read range used in various test setups was arbitrarily chosen to fulfil far-field criteria or whether that was the actual maximum read range. Furthermore, the use of differing interrogation/reception antennae and differing test setups also throws the read range metric into question.

In any case, the comparison drawn by the authors of Reference [11] is an important one and is certainly worth keeping in mind when reading this manuscript. Along with complementing the aforementioned paper, this manuscript sets out to review the current state-of-the-art in chipless RFID sensors. The Sensing section discusses many of the existing, unique sensor designs found in the literature and compares their measurement performance through the use of a table. This section also includes an overview of some of the materials used in the referenced sensor designs and other materials besides. This discussion on materials is aimed to complement the work of Karmakar et al. [15] but does not include experimental results relating to material performances.

### 1.2. Application Constraints

A useful RFID sensor for remote sensing is considered to be one that incorporates the following characteristics:Its response can be isolated and decoded even in a multi-sensor environment; therefore, some form of addressing should be supported.Has a read range of over one metre and should operate within the power limitations outlined by the relevant regulatory bodies.It can be fabricated in-situ using relatively simple fabrication techniques, e.g., using an inkjet printer. However, certain design approaches may be printable in the future, such as in the case of SAW tags and low-transistor count thin film transistor (TFT) tags so they will remain part of the review. Chipped RFID and related sensing is thoroughly reviewed in Reference [16] by Cui et al., and thus will not be further discussed here.Cost per unit is not of primary importance in this review and thus, basic thin film transistor tags and SAW tags are still reviewed. Although such a metric is of critical importance in the use of chipless RFID in barcode applications, this review aims to focus on the enabling technologies for chipless remote sensing.

### 1.3. Acronym List

RFIDRadio Frequency IdentificationSAWSurface Acoustic WaveTFTThin Film TransistorICIntegrated CircuitASKAmplitude Shift KeyingPSKPhase Shift KeyingUHFUltra-High FrequencySNRSignal-to-Noise RatioIDTInter-Digital ElectrodeRFRadio FrequencyPVDFPolyvinylidene DifluorideCRLHComposite Right/Left-HandedTLTransmission LineQPSKQuadrature Phase Shift KeyingPPMPulse Position ModulationDDLDistributed Delay LineIGZOIndium Gallium Zinc OxideADCAnalog to Digital ConverterMIWMagneto-Inductive WaveSIRStepped Impedance ResonatorOCOpen CircuitSCShort CircuitOOKOn-Off KeyingSRRSplit Ring ResonatorUWBUltra-Wide BandRCSRadar Cross-SectionLCInductor-CapacitorSiSiliconGOGraphene OxiderGOreduced Graphene OxidePEUTPolyetherurethaneELCElectric LCPVAPolyvinyl AlcoholSIWSubstrate Integrated WaveguideCNTCarbon NanotubeLDRLight-Dependent ResistorCdSCadmium SulphideFETField Effect TransistorSHMStructural Health MonitoringIDEInter-Digital ElectrodeMLAMeander Line AntennaPHEMAPolyhydroxyethylmethacrylatePMMAPoly(methyl methacrylate)PDMSPolydimethylsiloxanePEDOT:PSS poly(3,4-ethylenedioxythiophene)CABCellulose Acetate ButyrateMOSMetal Oxide SemiconductorSnO_2_Stannic(Tin(IV)) OxideZnOZinc OxideTiO_2_Titanium DioxideCOCarbon MonoxideCSRRComplementary Split Ring ResonatorBSTBarium Strontium TitanateOFETOrganic Field Effect TransistorPMOFETPressure-Modulated OFETrr-P3HTregioregular Poly(3-hexylthiophene-2,5-diyl)RHRight-HandedLHLeft-HandedPDPartial DischargeZImpedanceβPhase ConstantλWavelengthτGroup Delay

## 2. Radio Frequency Identification (RFID) Technology Overview

The commonly found passive RFID tag makes use of an Integrated Circuit (IC) to perform the necessary output signal modulation. These tags utilise bi-directional communication through a technique called “load modulation” [17,18]. Similar to a radar system, far-field RFID tags scatter the incoming signal back to the reader. This scattering takes place in two forms: antenna mode scattering and structural mode scattering [19,20]. The latter occurs due to currents flowing through the antenna, even under the condition that the antenna and load are matched. The former takes place due to reflections caused by load mismatches. Regardless of the matching performed, there is always a backscattered signal. Near-field tags are coupled to the reader antenna and the effect of load modulation has a similar effect, as described by transformer theory. This review is concerned with chipless RFID sensing, which in the majority of cases does not make use of load modulation, in the way that chipped RFID tags do. This section will briefly discuss load modulation, but an interested reader can find a more detailed coverage in the works of Finkenzeller [17] and the discussion presented by Rance [21].

Load modulation involves the alteration of the antenna load that results in a variation of the net backscattered signal. This alteration is controlled by the tag IC via a switch (i.e., a transistor as seen in Figure 1) and the switching frequency is usually much lower than that of the carrier [18]. Different loads can be used to achieve different modulation techniques, e.g., a resistive load can be used to achieve Amplitude Shift Keying (ASK), whereas a capacitive load can be used to achieve Phase Shift Keying (PSK).

### 2.1. Near-Field Technologies

This approach uses either inductive or capacitive coupling. This mechanism only operates in the near field of the antenna [17,18], which signifies the region before the electromagnetic field begins to separate from the antenna and begins to form a radiating wave, which occurs at a distance of *λ*/2π from the antenna [17]. In this region, the power falls off with distance^3^ and the tag and reader are coupled in a capacitive manner or more commonly, in an inductive manner [17].

The antennas used in this approach are usually loop antennas (for inductive coupling) and although many of the well-known examples of chipped RFID systems use near-field technologies, chipless near-field technologies have also been implemented. Examples of this include the work of Paredes et al. [22], that of Lee et al. [23], the sensor presented by Mukherjee in Reference [24] along with the design depicted in Reference [25] by Herrojo et al. The approaches used in these tags are basically the same as their far-field counterparts but with limited read ranges, and although this review concentrates on far-field chipless RFID systems, the same principles apply to near-field tag design. However, although the read range at the commonly used 13.56 MHz is several metres, the resonators to operate at this frequency are quite large. Therefore, many of the chipless near-field systems operate at ultra-high frequency (UHF) and microwave frequencies and thus have sub-metre read ranges.

Although it may seem that operation at the commonly used 13.56 MHz frequency may seem pointless, at this frequency several key environmental issues do not arise. The most important of these include the “water-metal” problem that UHF and microwave tags suffer from, a problem which is more thoroughly reviewed in Reference [18] by Dobkin.

### 2.2. Far-Field Technologies

This approach uses radiation as opposed to coupling mechanisms to communicate between the tag and the reader. In this region, power falls off with distance^2^ and by analysing the backscattered electromagnetic signal [17]. Similarly, load modulation techniques are used in this approach. The range of this technology is predominantly controlled by the signal-to-noise ratio (SNR) of the channel, the resonant characteristics of the tag and the free space loss. Unlike the near-field approach, the range of this technology is not limited to the same degree and ranges above 1 m are common. Most of the RFID tags that use this approach operate in the UHF and microwave region of the spectrum and thus, unlike near-field RFID tags, these tags suffer to a greater degree with unwanted interactions with water and metal.

### 2.3. Reader Architecture

One benefit with the use of chipped, or indeed TFT-based RFID tags, is that they can usually be interrogated with existing chipped RFID readers and in some cases, devices that support NFC, such as certain mobile phones. In order to make use of the various chipless RFID sensors that may exist in an environment, a new comprehensive reader system is required that will perform the following functions:Interrogate sensors individually, through the help of tag addressing schemes and perhaps with the help of localization techniques such as beam steering.Decode the received signal and remove or avoid unwanted environmental signals that may also be present.Determine the magnitude of the measured stimulus and report such finds to an appropriate display or to a larger system, such as a database server.Be capable of reading tags from an appropriate range and comply with relevant regulations.

Many of the works reviewed in this paper have been tested using equipment such as a VNA, which does not perform all of the functions outlined in the above points. In many cases, there are reference measurements performed using a calibration tag to allow for adequate subtraction from the signal received during actual testing [26,27]. A significant amount of research has been done and continues to be performed on designing suitable reader systems for the various chipless RFID tags that exist. The delay between scattering modes in transmission line-based RFID tags more readily allows for the use of existing Impulse Radio Ultra-Wideband (IR-UWB) RADAR technology to be used for time domain decoding of the response signal [27]. Such devices are already compliant with regulations and are relatively cheaper to buy or implement, as noted in Reference [28] by Ramos et al. and by Garbati et al. in both References [29,30]. Other popular interrogation systems include Frequency-Modulated Continuous Wave (FMCW) RADAR readers, which have been reviewed in Reference [27] for the purposes of performing the interrogation duties needed for spectral domain tags. However, the lack of compliance of this interrogation method with FCC UWB regulations is an issue that needs to be tackled for the overall technology to be adopted in real life scenarios [31]. Other improvements on this technology have been proposed, such as that by Forouzandeh and Karmakar [31].

The design of such a device requires significant consideration and is heavily affected by many of the characteristics of the RFID tag under interrogation. These tag characteristics include: polarization, presence of delay lines and characteristics of the tag resonator/antenna, such as composition materials and radiation/scattering efficiency. Although a thorough review on this topic is required to solidify the current state-of-the-art in this area, it is beyond the scope of this specific review to do so. For now, interested readers are pointed towards the works of Karmakar et al. [27] for a more in-depth discussion on the topic. This section is only introduced to highlight to the reader that the design decisions taken in RFID sensor design have a direct and influential effect on the complexity of the reader architecture and on its compliance with existing regulations.

### 2.4. Scope of this Review

This review will focus on chipless RFID tags and sensors which usually do not use the load-modulation communication technique. Although some of the reviewed sensor design approaches incorporate an IC, this element is not responsible for the sensing mechanism in the referenced works, and thus, the discussion presented around these works focusses on the resonator or antenna that acts as the stimulus-sensitive element. Similarly, basic transistor-based tag designs that have been printed using Thin Film Technology (TFT) methods are also included in this review because this technology is flexible in that the design footprint could be modified to include both addressing and sensing strategies as transistor-based sensing strategies is a well-established field.

## 3. Time and Phase Domain Tags

### 3.1. Surface Acoustic Wave (SAW) Tags

The topic of SAW-based sensing is a well-established field and has been well described by Gardner et al. [32], by Finkenzeller [17] and other dedicated books on the topic. Figure 2 depicts the basic configuration of a SAW tag, along with the interrogation and response signals for this device. This device includes an antenna that is connected to a metallic structure called an Inter-Digital Transducer (IDT). This planar structure is fabricated on a piezoelectric substrate using etching or “lift-off” techniques [32]. Upon reception of an appropriate signal, the electric field generated in the IDT results in a piezoelectric coupling which results in the generation of a surface acoustic wave [33]. A more in-depth discussion of this apparatus, IDT design and characteristics of the resulting waves are beyond the scope of this review and are thoroughly discussed by Gardner et al. [32] and in the discussion from Wilson and Akinson [34]. The resulting acoustic wave travels towards metallic reflectors that are present on the substrate and the wave is partially reflected back to the IDT and partially forwarded on to subsequent reflectors. The acoustic wave travels at a speed much lower than light, and thus, environmental echoes of the RF interrogation signal do not interfere with the newly formed RF signal returning from the SAW tag if sufficient space is left between the IDT and the first reflector. The minimum distance between successive reflectors is highly dependent on the incoming signal duration and the propagation velocity of the acoustic wave, and thus, SAW tags can also be made quite compact due to the slow velocity of acoustic signals.

Since SAW tags require a piezoelectric material such as Quartz as a substrate, they usually cannot be fabricated using deposition techniques [21]. However, certain types of ceramics and plastics, for example, polyvinylidene fluoride (PVDF), have exhibited piezoelectric characteristics [35]. Such materials may allow for in-situ fabrication of the tag, building on the work presented by Jones et al. [36], Ting et al. [37] along with the work by Tuukkanen and Rajala [38]. Further discussion on printable piezoelectric materials can be found in Reference [39].

### 3.2. Delay Line Tags

The topic of delay line RFID tags has been adequately summarised and reviewed by Herrojo et al. [11] and by Forouzandeh and Karmakar [13]. However, works of note will be briefly reviewed here and a further discussion will be presented on less known implementations of this approach.

A 4-bit time domain RFID tag is presented in Reference [40] by Chamarti and Varahramyan. This approach involves using two microstrip lines as transmission paths for the incoming signal, where one path is much longer than the other. The two paths can be joined (via an isolator) at integer durations of the input pulse on the longer path so that the main (straight) path now has two distinct pulses on it. The longer (usually meandered) path is terminated and the main path is connected to the output antenna. Impedance mismatches are used as a means of addressing by Zheng et al. [41]. This is implemented by capacitively loading a matched transmission line at various points. The inclusion of a capacitor will result in a reflection signal that would not be present otherwise. The capacitor values used in this tag need to be varied along the length of the line to keep the amplitude of the response bits in the output signal at a constant value. More recent uses of delay lines in RFID tags include a variety of ultra-wideband (UWB)-compliant tags, such as that outlined by Hu, Law and Dau [42]. Tags of this type have been successfully interrogated using Impulse Radio (IR) UWB RADAR systems, as seen in the work of Ramos et al. [28] and Lazaro et al. [43].

Composite Right-Left Hand (CRLH) transmission lines have been identified as a means to improve the bit-density of reflection-based chipless RFID tags [44,45]. With this approach, the group velocity ($v_{g}$) of the line can be engineered such that it is far lower than that found in regular transmission lines [13]. Equation (1), found in [45], describes the relationship between the length of the line ($L_{Line}$), the pulse width ($L_{Pulse}$) and the maximum encodable bits ($n_{bits}$). From this equation, it can be seen that by artificially reducing the effective group velocity, the bit-density can be increased.
(1)$$n_{bits\left(\max\right)}=\frac{L_{Line}}{L_{Pulse}}=\frac{L_{Line}}{v_{g}\left(\Delta f^{-1}\right)}\;where\;\Delta f=Pulse\;Bandwidth$$

CRLH transmission lines are metamaterial-inspired transmission lines that effectively exhibit metamaterial characteristics through the direct engineering of the impedance and dispersion/(phase constant) of the transmission line, as described by Martín in References [46,47] and by Caloz and Ioh [48]. The unit cell of the common right-hand (RH) transmission line is shown in Figure 3a and beside it in Figure 3b is the dual circuit, the left-hand (LH) transmission line. Implementing an LH transmission line brings with it parasitic RH elements, so the final result is coined a CRLH transmission line and can be modelled as the circuit presented in Figure 3c.

The circuit itself can be fabricated in a variety of ways, such as with planar inductor and capacitor elements or by using some negative-µ and/or negative-ε particles as loading elements for a regular transmission line. The extra circuit elements present in a CRLH transmission line (TL) circuit model allows for the tailoring of the (Bloch) Impedance, *Ζ* and Phase Constant, *β*. The relationships between these properties and the various CRLH elements are described in Reference [47].

As reviewed in Reference [13], this approach has been used in several RFID tag designs [44,45,49], where delay time was effectively doubled in one case and in another, a 28% reduction in footprint size was the result. Some issues have been highlighted in these papers regarding required bandwidth, dispersion effects and losses when using CRLH TLs to achieve compact delay lines. Many of the tags that use CRLH transmission lines, do so to achieve phase modulation, e.g., quadrature phase shift keying (QPSK). This is carried out by Mandel et al. [44] and by Chandrasekaran et al. [49], through the use of matching circuits or specific termination loads whose reflection has the desired phase response.

Other uses of this transmission line include RFID sensors that make use of the fact that LH resonant harmonics increase negatively with frequency and higher order harmonics in LH TLs can be made lower than their RH counterparts. A review and subsequent use of these effects can be found in the works of Penirschke et al. [50] and that of Puentes et al. [51].

### 3.3. Group Delay-Based Tags

An alternative conceptual design for a time domain-based RFID tag was put forward in 2010 by Gupta et al. that made use of microwave C-section elements [52]. This tag used group delay engineering (see next paragraph) to achieve pulse position modulation (PPM), as the delay induced by the structure seen in Figure 4 is frequency-dependent. Therefore, a wide frequency sweep can be used to interrogate such a tag and determine the encoded address.

The structure described above, referred to as a “C-section”, is a Dispersive Delay Line (DDL) structure that can be used to introduce a group delay at a specific input frequency. It can be considered to be one such way to implement a planar bridged-T equaliser [53]. These C-section structures are considered to have all-pass filtering characteristics and can perform equalisation operations. A diagram of the DDL in question can be seen in Figure 5. In essence, this structure can be considered to be a transmission line section of length “l” that has a short circuit (S.C.) load attached [53,54]. Given the operation of a *λ*/4 transformer, this short circuit will be converted into an open circuit when excited by a wave with a wavelength of $\lambda_{O.C.}=\frac{4l}{2m+1},$ where, *m* is an integer. The open circuit condition will decay symmetrically about the open circuit frequency and normal transmission line behaviour occurs between successive open circuit frequencies. Since this structure is not just a single stub, a coupling capacitance exists between the two branches and this provides a secondary signal path [55,56]. Equation (2), found in Reference [53], describes the spectral response of this circuit. The group delay, which is the derivative of the angle of Equation (2), also found in Reference [53], is presented in Equation (3).
(2)$$S_{21}\left(\theta \right)=\frac{\sqrt{1+k}\;\cos\theta -j\sqrt{1-k}\;\sin\theta }{\sqrt{1+k}\;\cos\theta +j\sqrt{1-k}\;\sin\theta }\;where\;\theta =\;\beta l=\;\frac{2\pi l}{\lambda }$$
(3)$$\tau \left(\theta \right)=-\frac{d\phi S_{21}\left(\omega \right)}{d\omega }=\frac{2a}{a^{2}+\left(1-a^{2}\right)\cos^{2}\theta }\frac{d\theta }{d\omega }\;where\;a=\;\sqrt{\frac{1-k}{1+k}}$$

The factor, “*k*” is referred to as the coupling factor/coefficient and is derived from directive coupler theory. For the sake of this discussion, it will be said that it is heavily related to the coupling capacitance between the two stubs [53,55]. This dependence drives the magnitude of the group delay induced by the structure and the magnitude of the spectral response of the system. Due to this dependence, sections that have longer lengths, e.g., have larger values of “*m*”, will exhibit greater levels of group delay [53,54]. This can be understood simply by considering the fact that the coupling capacitance is now larger. These sections can also be cascaded together as a means to achieving larger group delays.

### 3.4. Transistor-Based Approaches

Thin film transistors (TFTs) made out of Indium Gallium Zinc Oxide (IGZO) and other such materials, have been investigated for developing printable RFID technology. Many examples of TFT-based RFID tags have already been developed, such as the works of Myny et al. [57,58,59], along with the designs outlined by Heremans et al. [60] and Cantatore et al. [61]. One 8-bit example designed for inkjet printing is presented by Kjellander et al. [62], that incorporates 300 TFTs in a surface area of 34 mm^2^.

The sensing possibilities with thin film transistor (TFT) circuits is quite vast as analog to digital converters (ADCs) can be readily implemented, as discussed more thoroughly by Ussmueller et al. [63]. Besides the use of external sensing components, transistor technology can be used in a variety of sensing applications including: X-ray detection, as described by Moy [64], biosensor applications, such as that developed by Magliulo et al. [65] and temperature, as reviewed by Hagelauer et al. [3]. This section is not a thorough review of the possible sensing possibilities of TFT RFID systems but attempts to outline how this technology could be used as remote sensors. The later discussion outlines how resistive and capacitive sensors can be implemented with these devices. This is important as the materials to be reviewed in the latter stages of this paper are all used in either a resistive and/or capacitive sensor configuration.

### 3.5. Tabular Comparison of Time Domain Approaches

Below is a table (Table 1) comparing the various time domain RFID tags presented above. Not all of the columns are populated, either due to parameter unsuitability or lack of information. The information in this table has been presented so as to critically compare the various tags in this section. The advantages and disadvantages are based on the information presented in the paper under review and do not include further analysis. The metrics chosen include tag size and number of bits encoded. Where necessary, the tag dimensions have been estimated based on the content of the paper and tags which do not include an antenna or other features have been appropriately flagged. The operating frequency of the tag has also been recorded as this introduces some measure of how susceptible a tag will be to the “water-metal” problem. Likewise, the interrogation pulse type has also been included as it to some degree indicates the complexity of the reader system needed to interrogate the tag in question. The polarization of the resulting tag has not been included in the following Table, as the topic relating to antenna design has been largely left out of RFID tag designs of this sort.

## 4. Spectral Domain Tags

### 4.1. Filter-Based Addressing

The approaches outlined in this section use similar antenna setups to those outlined earlier in the time domain examples. These include both single antenna and dual antenna configurations that use either a reflecting termination load or use separate transmit and receive antennas, respectively.

Although there are many ways to encode addresses into response signals, one simple way to do so is through the use of Stepped Impedance Resonators (SIRs). The design of an SIR involves cascading alternating microstrip elements with high and low impedances together. Using simplified analysis of short transmission lines, these elements are representative of inductors and capacitors, respectively [56]. Quarter wavelength SIRs are set as having that specific length and unlike regular SIRs, which connect the input and output together, one end of these SIRs are left in an Open Circuit (O.C.) or S.C. condition [78]. Through the principle of the *λ*/4 transformer, the SIR will transform the O.C. condition to a S.C. condition or the alternative arrangement. This SIR, as seen in Reference [79], in theory will result in a S.C. response at the frequency of interest. The design of such a filter is described thoroughly by Sagawa et al. [80] and by Lin et al. [81].

Amin et al. developed a Partial Discharge (PD) RFID sensor utilising several coplanar, Tri-Step *λ*/4 SIRs to alter the spectrum of an incoming UWB partial discharge signal [79]. The Partial Discharge (PD) RFID sensor developed by Amin et al. utilises several coplanar, Tri-Step *λ*/4 SIRs to alter the spectrum of an incoming UWB partial discharge signal [79]. The transmission of the resulting signal could then allow for easy localisation of the PD signal. Other noteworthy RFID-based examples of SIR usage include those outlined in the works by Nijas et al. in both References [82,83] along with the work of Sakai et al. [84]. Although cascaded SIR designs can be quite large, this issue could be mitigated to some degree through the use of innovative SIR layouts built on the works of Lin et al. [81], Velázquez-Ahumada et al. [85], Pal et al. [86] and Mokhtaari et al. [87].

Alternative filter-based approaches include loading a transmission line with other elements such as resonators or stubs. The resonant response of a nearby resonator attenuates the transmission of the signal through the transmission line; likewise, the use of open-circuit *λ*/4 stubs allows for strong reflection at the desired frequency. Example tags that use spiral resonators as loading elements include that outlined by Preradovic et al. in References [5,88]. Similarly, stubs were used as loading elements in Reference [89] by Prabavathi et al. and by Mousa et al. [90], along with that found in Reference [91] by Nijas et al. and the work of Khaliel et al. presented in Reference [92].

### 4.2. Resonator-Based Tags

Similar to the transmission line approaches discussed above, resonators can be coated or modified in other ways to create a sensing resonator. This section will focus on the chipless RFID sensing solutions that make sole use of resonant elements and whose resonant frequency changes in accordance with the sensing stimulus.

#### 4.2.1. Dipole-Inspired Resonators

Resonators such as the hairpin or C-shaped resonator described by Lee in Reference [78] have been extensively used in passive RFID tags. The basic form of this resonator is just a simple folded version of the dipole resonator; however, as the distance between the two prongs of the resonator decreases, the distributed capacitance between these two elements begins to influence the resonant characteristics of the resonator. Both implementations of this resonator are diagrammatically represented in Figure 6a. Islam et al. [93], Figueiredo et al. [94] and Mumtaz et al. [95] all used this form of resonator in their tag design. The implementation described in Reference [93] uses slots of this type with a large distance between the prongs and partially back-fills in the ends of the slots with a metallic filling as a means of addressing. This can be done during or after the default resonator is fabricated. In either case, this allows the resonant frequency of the resonator to be set so as to encode a specific address using ASK encoding. The implementations by Mumtaz et al. [95] and Polivka et al. [96] uses resonators of differing lengths to encode several address bits; however, this approach uses On-Off Keying (OOK) encoding as it represents zero by shorting the prongs of the desired resonator. One final example of the use of this structure can be found in Reference [97], where it is used to achieve both frequency and phase encoding. The frequency encoding was performed similar to that found in Reference [95], and unlike the implementation in Reference [93], the gap between the prongs was small enough to become significant to the resonant response. This gap was varied in the tag implementation by Vena et al. [97] to achieve a further degree of modulation as the Q-factor of this resonator shifts significantly with changes in gap length.

Other simpler resonators such as slots, L-resonators and regular dipole resonators have also been used for chipless RFID addressing schemes. Similar to those hairpin resonators discussed above, such implementations could support both OOK and ASK encoding schemes. Interesting implementations of this resonator include that by Nair et al. [99], the work of Deng et al. [100] and also the tag presented by Sharma et al. [101]. Other works of note include the tag developed by Jalaly and Robertson [102], which uses dipole resonators, and the work of Fan et al. [103], which uses L-shaped resonators.

Also, the principle of the hairpin resonator can be used to modify existing resonators, by cutting out similarly sized slots. As described by Xu and Huang [98] and depicted in Figure 6b above, cutting slots whose equivalent hairpin length is *λ*/4 leads to extra resonant characteristics of the resulting resonator. This approach was also taken in the RFID tag presented by Anam et al. [104]. One final variant of this approach was outlined by Khaliel et al. [105], which made use of cross-dipole resonators.

#### 4.2.2. Ring Resonators

Ring resonators can also be used as a means of RFID-based address encoding. One useful characteristic of such an approach is that polarization losses caused by the misalignment of the linearly polarised reader signal with the tag do not arise with this structure. Impressively, an 8-bit tag was implemented by Islam et al. within a 1.5 cm^2^ footprint in Reference [106], as seen in Figure 7. The resonant frequency of ring resonator structures can be approximated as that in Equation (4), taken from Reference [107]. However, several slightly different interpretations have been discussed in papers such as in References [108,109,110]. It is believed that many of the harmonics are not present in real-world implementations of slot ring resonators due to characteristics of the slot width, as described by Abbas et al. [109]. In any case, the main aspect of this discussion is that the structure is resonant when the wavelength of the incoming signal is some multiple of the electrical length of the circumference. Other RFID-based implementations of interest include that outlined by Vena et al. [108], along with that used by Fawky et al. [111] and the tag discussed by Yang et al. [112]. More recently, Abbas et al. also makes use of this resonator type in Reference [108] and uses redundant tags in order to improve the robustness of the radar cross-section measurement of the complete tag.
(4)$$f_{n}\propto \;\frac{nc}{\pi r\sqrt{\varepsilon_{eff}}}\;where\;r=ring\;radius\;and\;n=1,\;2,\;3,\ldots$$

Eccentric ring resonators have also been used to make dual-band antennas for RFID tags, as outlined by Barman et al. [113] and ring resonators have also been used in a variety of sensor implementations including that proposed by Singh et al. [114]. Loop resonators have also been used for RFID addressing. These resonators have similar geometric properties to ring resonators but only one of their two edges form a ring shape. An example of its implementation in chipless RFID tag designs include the tag outlined by Khan et al. [115].

Similar to ring resonators, Split Ring Resonators (SRRs) are used extensively in chipless RFID systems. Proposed by Pendry [116], the use of such resonators has certain advantages over conventional ring resonators. The main advantage is their compact size as the distributed capacitance in the structure gives rise to a very low resonant frequency. The performance of these resonators in comparison to similar ring resonators is discussed by Aydin et al. [117] and will not be further discussed here. They have been used as resonator elements in RFID tags in Reference [118], which was proposed by Jang et al. and as antennas, like that outlined by John et al. [119]. The implementation in Reference [118] used several such resonators to make a composite RFID tag. A similar approach was also taken by Shao et al. with a different resonator in Reference [120].

#### 4.2.3. Space Filling Curves

Numerous papers have utilised space-filling curves for antenna, resonator and microwave filter design. These fractal-type structures boast a much smaller resonant frequency than their equivalently sized monopole/line structures [121,122]. Examples of such shapes can be seen in Figure 8 below. Since their characteristics are similar to that of a dipole, further discussion of their resonant performance will not be presented here. However, it is discussed in depth in by González-Arbesú et al. [121], by Jarry and Beneat [122], and by Lee [78] and should be reviewed when comparing their performance to that of a regular dipole antenna. The alternative approach, which is to use a dipole resonator, has also been implemented in chipless RFID systems, such as that described by Nair et al. [99], but utilising larger amounts of real-estate.

Such curves can be used in a variety of ways for UHF RFID tags. One such example proposed by McVay et al., can be found in Reference [123], where several curves of differing resonant frequencies are used to form an n-bit spectral ID. Alternative RFID-based uses include directive, regular/multi-resonant antennas, as proposed by Alibakhshi-Kenari et al. [124] and by Murad et al. [125]. The use of space filling curves as High Impedance Ground Planes (HIGPs) to enhance antenna directivity has also been discussed by McVay et al. [123].

### 4.3. Tabular Comparison of Spectral Approaches

Below is a table (Table 2) comparing the various spectral RFID tags that were reviewed as part of this research. Some of these tags are not included in the above discussion, but their underlying operation has been succinctly described by Herrojo et al. [11]. Along with the metrics used in Table 1 and Table 2 includes other metrics of interest for spectral domain tags. These include operating frequency and frequency range along with polarization. From the recent literature, it appears that more has been done in the area of spectral domain tags to deal with polarization issues, and thus, it is important to recognise the efforts in some of these works to overcome this problem. Other parameters of interest include the maximum and minimum power difference in S-parameters in a spectral domain tag when a zero bit is encoded as opposed to a one. This is related to many characteristics, including operating frequency, resonator design, degree of coupling (if applicable) and other factors linked to materials and fabrication effects. Regardless of this, the minimum variation gives an indication of the sort of channel characteristics this tag design can endure and both metrics define important characteristics of the reader design requirements.

## 5. RFID Sensor Implementations

### 5.1. Volatile Organic Compound (VOC) and Humidity Sensing

TFT-based gas sensors have also been implemented that use stimulus-sensitive gate coatings, examples of this include the work of Liao et al. [131] and that of Barker et al. [132]. The former reports that coating the gate can lead to a variety of sensible effects including changes in TFT transconductance, threshold voltage, series resistance and leakage current. The latter sensor relies solely on impedance dependence of the coated film to modulate the turn-on time of the transistor. Other approaches, such as that outlined by Li and Lambeth [133], rely on variations of the electrical characteristics of the grain boundaries when exposed to Volatile Organic Compounds (VOCs). More information on the topic of TFT gas sensing can be found in Reference [2].

Many SAW-based gas sensors have also been developed, as discussed in Reference [4] by Jakubik and in Reference [134] by Rodríguez-madrid et al. Their operation involves applying a sensitive coating to the substrate that absorbs the target molecule, and this causes a change in the boundary conditions for the acoustic wave which results in an attenuation of the signal and/or in the wave velocity [4]. Example coatings include: PVA for humidity sensing, as seen in Reference [135] by Chen and Kao, along with the work of Penza et al., which uses CNTs for gas sensing [136].

The group delay characteristics of the C-section/hairpin resonator are exploited by Perret et al. [7] and Nair et al. [75] for the purposes of humidity sensing. These implementations deposit silicon (Si) nanowires on the resonator to allow for the shorting of the resonator at a shorter distance along its length. As discussed later, Si Nanowires respond to increases in humidity with a proportional decrease in impedance. The latter reference also makes use of spectral domain techniques for humidity detection, as the permittivity of the nanowire deposit changes due to the adsorption of water.

One example of the use of resonators in RFID sensing can be found in the work of Fan et al., presented in Reference [103]. This tag utilises a symmetric design with several L-shaped dipole resonators that are used for both sensing and addressing. The centre region and the nearest resonator set are coated with PVA to achieve a humidity sensing RFID tag. Similarly, a coated spiral antenna coated in PEUT is used by Potyrailo and Surman [137] for humidity sensing. Other gas and humidity sensors include the work of Bogner et al. [138], which uses circular ring resonators along with zeolite to create humidity/ammonia sensor. An inter-digital electrode (IDE) is attached to an antenna by Ren et al. [139] for remote sensing purposes. This tag uses graphene oxide (GO) as the sensing film. The resulting sensor can be used in temperature and humidity sensing applications. A humidity sensor outlined by Deng et al. [100] makes use of slot resonators for addressing and sensing, as one resonator was coated with Si nanowires. As discussed earlier, the absorption of moisture by the nanowires leads to a permittivity change which results in a frequency encoded humidity sensor. A similar approach is taken in Reference [140] by Vena et al. that uses loop resonators coated with Si nanowires for the sensing of humidity. Other implementations of interest include the multi-parameter sensor outlined by Amin et al. [127], which makes use of two ELC resonators and a set of hairpin/C-section slot resonators, the latter of which are used for addressing, and one of the other resonators was coated with PVA for humidity sensing. The electric LC resonator, as shown in Figure 9, is a negative-ε structure which has fewer unwanted self-coupling effects than its SRR-based equivalent [141]. It consists of two inductive branches, separated by a central capacitive element, and this symmetry results in less cross-polarization effects taking place. The resonant frequency $\left(\omega_{0}\right)$, is given as $\sqrt{2/LC}$ and is highly sensitive to changes in capacitance, as the fundamental mode is generated by electric coupling to the central capacitor [46,141]. A more thorough discussion on this particle can be found in Reference [141] by Withayachumnankul et al. and by Martín in Reference [46]. A design strategy for such particles is presented in Reference [142] by Pushkar et al. and in Reference [143] by Schurig et al.

Chen et al. developed an SRR-loaded Substrate Integrated Waveguide (SIW) in Reference [144], which when coated with Silicon nanowires, resulted in a spectral domain humidity sensor.

Transmission line-based solutions have also been used for spectral domain RFID sensors. One example of interest is that outlined by Dominic et al. [145]. This implementation utilises two antennas joined via a loaded transmission line. Spiral resonators are used as loading elements and are used for both addressing and sensing, as the final resonator is coated with PVA to allow for a temperature-dependent resonant frequency of that spiral.

Other RFID sensors make use of the substrate material as the sensing material, including the work of Wiwatcharagoses et al. [146]. The design in the referenced paper relies on capillary condensation within the substrate to allow for capacitive-based sensing of volatile gasses. Other RFID gas sensors of note include those outlined by Yang et al. [147] and Ong et al. [148], which make use of carbon nanotubes (CNTs).

### 5.2. Velocity and Direct Permittivity Monitoring

CRLH-based transmission lines have been used for contactless permittivity and velocity detection in the works of Puentes et al. [51] and that of Penirschke et al. [50]. The low-frequency, compact tag found in Reference [127] is a CRLH transmission line that can be used as a capacitive-type sensor. As described earlier, such a transmission line can be built out of regular inductors and capacitors, and in this case, utilises the large planar capacitors as a region for permittivity monitoring. Velocity changes are detected via the change in permittivity of the external material and move across the capacitive region and onwards. A similar approach is taken in Reference [50] for material flow sensing in a pipe.

The tag presented by Xia and Wang [149] makes use of a bowtie antenna structure connected to a SRR. The split capacitance is used as a mounting point for a capacitive-based sensor to be added. This approach is taken by Sarabandi and Li [110] and is also taken in Reference [150], where Ramaccia et al. use it for the remote measurement of moisture levels in soil.

A finger contact sensor has been outlined in Reference [151] by Kim et al., that uses two resonators. One of which has a meandered element that when touched by a finger results in that resonator having a different resonant frequency.

Costa et al. take a different approach in their development of a permittivity sensor in Reference [152]. This tag makes use of two dipole-like resonators orientated at 45° relative to the incident electric field component. Using a cross-polar antenna setup with the reader system, the large reflection signal from the Material Under Test (MUT) is avoided, and the information of interest is determined from the variation in resonant frequency of the sensor. A similar approach is taken by Lázaro et al. [153], where a bent dipole is used to achieve a similar effect. This work also utilises several of such elements to strengthen the response signal and thus improve the overall read range of the sensor system. Other approaches make use of stub-loaded transmission lines, whose capacitance is dependent on the MUT present. An example of this design can be seen in the work of Girbau et al. [154].

A time domain sensing approach is taken by Ramos et al. [155]. This tag embeds a meandered transmission line in the MUT, which is concrete in this case and relies on the fact that propagation velocity in the line is dependent on the permittivity of the MUT. The resulting antenna mode signal is thus delayed sufficiently and in proportion to the permittivity of the MUT.

### 5.3. Ambient Light Intensity Monitoring

A noteworthy implementation of a transmission line-based sensor can be found in the work by Admin et al. presented in Reference [156]. This tag uses *λ*/4 SIRs for both sensing and addressing. The resonators used in Reference [156] are open-circuit, but a light-dependent resistor (LDR) is used to short one of the resonators to ground. The LDR uses a material called Cadmium Sulphide (CdS), whose impedance is dependent on ambient light intensity, thus the resonant frequency of this SIR is altered.

### 5.4. Rotation/Angular Displacement Sensing

The sensor outlined by Genovesi et al. [157] uses two stub-loaded resonators for rotation measurement. One resonator is attached to the rotating element and the cross-polarization reflection is used as a means of measuring rotation angle. Other rotation monitoring sensors of note include the work of Matbouly et al. [158]. This approach uses a slot resonator and a linearly polarized reader antenna to make use of rotation-dependent polarization mismatches between the reader and the tag.

### 5.5. Structural Health and Pressure Monitoring

Popular uses of SAW sensing include tire pressure monitoring, such as those outlined by Buff et al. in References [159,160], along with the works of Pohl et al. [161], Scherr et al. [162] and Dixen et al. [163]. Such approaches make use of phase encoding caused by the bending of the SAW substrate [161], through stress-induced changes in the piezoelectric material, resulting in a modulation of the SAW velocity [10,159,160,162]. More recent implementations, such as that by Humphries and Malocha [10], also include a robust OFC addressing scheme along with strain sensing abilities.

A variety of FET-based pressure sensors have been reviewed by Elkington et al. [2] and by Lai et al. [164]. An implementation of interest found in Reference [164] relies on the use of a piezoelectric polymer coating that is deposited on the gate to act as a means of modulating the channel current. An example of its implementation can be found in the work of Adami et al., presented in Reference [165]. Other implementations discussed in Reference [164] make use of a pressure-sensitive charged capacitance applied to the gate. Upon exposure to external pressure, the capacitance varies, which also leads to a modulation of the output current [164].

Crack detecting structural health monitoring (SHM) sensor tags were proposed by Nappi and Marrocco [166] and Zhang et al. [167]. The former tag uses a space filling curve-based skin for the structure of interest. Tests involving the fracturing of this curve result in a 4 MHz change in resonant frequency. The tag found in Reference [167] detects surface cracks in an aluminium substrate by positioning the cavity of the antenna above the crack, as the current in the antenna is highest there. The presence of a crack alters the field distribution and leads to an increase of the electrical length of the antenna. Another resonant-based strain sensor of note was outlined by Alipour et al. [168], that uses a flexible Kapton backing and layered LC resonator. The resulting sensitivity was estimated at −0.2 MHz/ε_r_ and needed 100 µm fabrication tolerances.

There are several other ways to implement an RFID strain sensor. For example, the sensor outlined by Daliri et al. [9] is a simple circular resonator that, when deformed, has a different radar cross-section. Another example, outlined by Jatlaoui et al. [169], uses a high-resistivity silicon layer that is suspended above a planar resonator. When pressure is applied to the upper silicon layer, it deforms/bends and alters the permittivity experienced by the resonator, which in turn causes a change in resonant frequency for the device.

Another interesting SRR-based RFID sensor is that outlined by Melik et al. [170]. This strain sensor based on an SRR grid array design supports up to 23,100 µε. Other tags rely on the permittivity of the substrate as an indication of its structural health. One example of this is the tag described by Suwalak et al. [171]. This tag is used for concrete monitoring as the resonant peak in the radar cross-section is, in this case, indicative of the health of the structure.

The sensor presented in Reference [172] by Kim et al. uses an LC resonator whose inductance and capacitance varies with applied strain. It comprises of a single-loop inductive element and an IDE-style capacitive element, which can be altered to allow for unique tag addresses. A diagram of the tag is presented below in Figure 10 and the equations describing the geometric dependence of the resonant frequency can be found in References [173,174]. The referenced paper uses a silver nano-ink and a direct stamping method to fabricate the above sensor. The properties of the resulting sensor support strain levels up to 8% with sensitivity levels of between 15 MHz/%ε and 52 MHz/%ε for strain levels below 4%.

Other sensing techniques of note include the RFID sensor presented by Occhiuzi et al. [175], which modifies its radar cross-section (RCS) in response to mechanical strain. This is achieved using a Meander Line Antenna (MLA), as seen in Figure 11. When stretched, the impedance of the antenna varies quite significantly as the electrical length and degree of coupling between antiparallel elements decreases. Although this tag incorporates an IC, the sensing element is purely based on the change of antenna impedance. In its initial state, the antenna is conjugately matched to the IC using a T-matching circuit, but changes in the antenna impedance will result in a backscattered signal. The resistance of the antenna grows initially, as described by Best [176], and decays off in a similar way until the effects of a heavily meandered antenna disappear. In a similar fashion, the reactance of the antenna changes considerably as the meandering effects dissipate because the changing inductive and capacitive characteristics of the nodes (corners) and nearby elements. Mathematical modelling of such structures is outlined in Reference [177] by Das et al.

The prototype testing of this tag design involved implementing such a tag with overall dimensions, *h* and *w,* both having 36 mm dimensions. The MLA was fabricated out of metallic wire and was manually soldered onto the T-matched circuit, which was etched out on FR4 fibreglass board. Elongation of the device involved tying plastic wires around the extreme ends of the MLA. The RFID reader consisted of an interrogation antenna with directivity of 3.3 dB, mounted 60 cm from the tag and an interrogation signal with a strength of 20 dBm. Some of the interesting results of this paper include:The tag itself is only capable of withstanding up to 6% strain before plastic deformation.The sensitivity of the device can be approximated as 0.429 dB per %ε.

A similar approach is taken by Teng et al. [178], where a meander line antenna is used for the purposes of strain sensing. This tag makes use of a flexible substrate and supports compressive along with tensile loading. Interestingly, microfluidic fabrication techniques are used to fabricate this device and the resulting sensor supports up to 50% strain.

### 5.6. Temperature Sensing

Interesting examples of SAW tags include its use in remote sensing solutions, such as that described by Malocha et al. [179]. This example makes use of Orthogonal Frequency Coding and relies on the temperature-dependence of the SAW velocity of the piezoelectric, YZ lithium niobate substrate, as described in Reference [160].

The group delay tag outlined in the works of Elmatbouly et al. [180] also have temperature sensing capabilities. The work does not use sensitive coating however, but instead relies on the temperature-dependent permittivity of the PCB material for sensing purposes.

A different approach outlined by Mandel et al. [181] made use of a CRLH transmission line with a temperature-sensitive element. This material, barium-strontium-titanate, is used to create temperature-dependent capacitances to allow for phase encoding of the reflected signal. Other sensor implementations exist that make use of CRLH transmission lines, such as that devised by Saghati et al. [182], however they are not yet used in remote sensing applications.

An ELC resonator is coated with Phenanthrene by Amin et al. [127] to complete their multi-parameter sensor configuration. In this material, a permanent change in relative permittivity occurs at this sublimation material’s transition temperature (72 °C) [15]. Similar materials exist with different transition temperatures, as described in Reference [15].

A different approach is taken by Preradovic et al. [5], where a spiral-loaded transmission line is used with a thermistor as its termination load. This tag uses only a single antenna and the reflection coefficient of the tag is varied by the temperature-controlled impedance of the thermistor.

An RFID sensor of interest outlined by That et al. [183] makes use of an SRR which uses a bi-metallic strip that modulates the split capacitance through temperature deformation, which results in a reduction of the gap distance. The diagrams in Figure 12 describe the layout of this sensor. The capacitive sensing characteristics of SRRs have also been made use of in other RFID temperature sensors, such as that developed by Bhattacharyya et al. [184].

Other time domain-based temperature sensors include the work of Girbau et al. [185]. This tag uses a meandered transmission line with a temperature-dependent load element as its termination. Like other terminated delay line tags, the target stimulus is encoded in the magnitude of the antenna mode in the scattering response. This approach is also used in the temperature sensor outlined by Lazaro et al. [186].

## 6. Overview of Previously Used RFID Sensor Materials

A useful review and comparison of current and possible sensor materials is presented in Reference [15]. This section sets out to complement that work by expanding on the sensing mechanisms and to discuss further materials of interest.

### 6.1. Stimulus-Sensitive Polymer Films

Polymer films are among the most-used sensitive film in the reviewed RFID sensing solutions. These films are usually used in the sensing of humidity and in the sensing of more general chemicals such as ammonia or methane. An exhaustive review of polymer-based sensing is not the main objective of this review and thus readers will be given suggested links to more thorough literature on the topic. Many of the polymers that are briefly discussed require specific preparation methods to enable them to be deposited onto an RFID tag. These methods vary quite significantly between polymers and more details on specific preparation methods can be found in the referenced literature. Below is a brief discussion on some of the commonly used polymer films in RFID sensing.

#### 6.1.1. Polyamide

A polyamide film (Stanyl® TE200F6) was used by Amin and Karmakar [187] for UHF RFID-based temperature sensing. This polymer, polyamide/nylon 46, was deposited on one of the several spiral resonators that were used to load the transmission line of the tag. The mechanism by which this occurs is down to temperature-dependence of the dielectric relaxation time of the material. A thorough discussion on this property and its temperature-dependence by Kao can be found in Reference [188]. The result of using this polymer as a sensitive coating in Reference [187] led to a temperature sensitivity of 1.25 °C/MHz between 0 and 25 °C. An experimental evaluation of this temperature-dependent phenomenon in Stanyl is presented by Pawlikowski [189].

#### 6.1.2. Kapton

Kapton (C_12_H_12_N_2_O) polyimide is used in humidity sensing applications, including the works of Rivadeneyra et al. [190], Shibata et al. [191] and the work of Boudaden et al. [192]. Kapton has also been used in UHF RFID applications by Virtanen et al. in References [193,194]. Like all polyimides, Kapton undergoes hydrolysis when exposed to moisture or heat. This is the exact opposite to the formation process of polyimides, which involves a polycondensation reaction [195]. The interaction of water with Kapton is described by Ralston et al. [195] and results in a change of the polarizability of the polymer and thus a permittivity change of the material.

#### 6.1.3. Polyvinyl Alcohol (PVA)

This polymer has been used for capacitive-based humidity sensing by Chen and Kao [135] and by Amin et al. in References [196,197]. Polyvinyl Alcohol (PVA) is a hydrophilic polymer that is less sensitive to humidity than other polymers, such as polymethyl methacrylate (PMMA) or poly (2-hydroxyethyl methacrylate) (PHEMA) [198], but more sensitive than Kapton [196]. Interestingly, this polymer has been found to be less affected by the presence of other gasses, such as CO and NH_3_ [127]. A more thorough discussion of PVA and its applicability as a humidity sensing material can be found in the work of Karmakar et al. [15].

Although the conductivity of PVA increases due to water sorption [199], this polymer can be used as a capacitive-type sensing film for UHF and microwave sensor systems, as seen in Reference [200] by Amin et al. and Reference [127] by Amin et al. As with many of the polymers discussed in this section, PVA can be used as a standalone, sensitive coating, as seen in References [196,197] by Amin et al., along with the work of Chen and Kao [135]. Alternatively, PVA can be used as part of a composite sensing material, as seen in the work of Chatzandroulis et al. [198], along with the work of Andreev et al. [199] and Deshkulkarni et al. [201].

#### 6.1.4. Poly (2-Hydroxyethyl Methacrylate) (PHEMA)

Poly (2-hydroxyethyl methacrylate) (PHEMA) is used as a humidity-sensitive coating by Reddy et al. [202]. PHEMA is a hydrophilic polymer which swells as it readily sorbs moisture from the atmosphere [203,204]. The sensitivity of this polymer to humidity in comparison to other hydrophilic materials like PMMA, PVA and PDMS is researched by Chatzandroulis [198]. However, from that research, it was concluded that PHEMA is far more sensitive, due to the presence of the OH group to readily allow hydrogen bonds with H_2_O [205]. Since the permittivity of PHEMA and other such polymers depends on the polarizability of the inner macromolecular chains and on the free space present in the matrix structure [206], exposure to moisture causes a significant change in permittivity.

#### 6.1.5. Polydimethylsiloxane (PDMS)

PDMS (Polydimethylsiloxane) has been outlined as a viable sensing polymer by Harsányi [204]. A cross-linked variant of PDMS was used in Reference [207] by Hillier et al. for UHF RFID sensing of pH. Other sensing applications also exist for PDMS, including pressure sensing up to 1 MPa, as seen in the work of Lei et al. [208]. Both of these sensor implementations are capacitive in nature and thus are quite suited to spectral-based RFID sensing approaches. Other sensing uses of this material involve its inclusion with graphene or with nanomaterials for the creation of robust and highly sensitive strain gauges, as seen in the work of Chen et al. [209].

#### 6.1.6. Polyetherurethane (PEUT)

PEUT (Polyetherurethane) is a polymer with a glass transition temperature that is below room temperature, as described by Potyrailo and Surman [137]. An example of its use in sensing relative humidity and temperature levels can be seen in Reference [137]. At room temperature, the thickness of the PEUT film expands upon exposure to humidity, similar to other polymers like PAA (Polyacrylic Acid) and PVP (Polyvinylpyrrolidone), as described by Altenberend et al. [210]. The effect of polymer swelling only accounts for part of the permittivity change, as the permittivity of the measured vapour also contributes to the net permittivity change [137,179].

#### 6.1.7. Polymethyl Methacrylate (PMMA)

Polymethyl methacrylate (PMMA) is another polymer that can be used as a humidity-sensitive coating or for the sensing of chemicals, such as methanol, as reviewed by Harsányi [204]. The gas sensing properties of PMMA are broadly discussed by Korotcenkov [211] and the hydrophilic, gas sensing properties of PMMA are described succinctly by Matsuguchi et al. [212]. An example of the implementation of a PMMA-based gas sensor can be found in the work of Zhang et al. [213].

#### 6.1.8. Poly(3,4-Ethylenedioxythiophene) Polystyrene Sulfonate (PEDOT:PSS)

PEDOT:PSS (poly(3,4-ethylenedioxythiophene) polystyrene sulfonate) has been used in numerous sensor implementations [214,215] for temperature and humidity sensing. PEDOT can also be used for conductivity-based pH sensing and for sensing other stimuli, as described by Amin et al. [200] and Bernards et al. [216]. From the literature reviewed by this author, this material is also used as part of a composite sensing film for the sensing of stimuli other than pH and humidity by Vena et al. [217] and has been outlined as a candidate material for printable, flexible electronics by Cui et al. [218].

#### 6.1.9. Cellulose

Like many other polymers, such as PMMA and PHEMA, cellulose and some of its derivatives can be used in a variety of chemical sensors, such as humidity, ion and gas detectors, as seen in the work of Harsányi [204] and that of Matsuguchi et al. [212]. Cellulose nanofiber sheets are used for the sensing of humidity by Eyebe et al. [219], whereas functionalised cellulose acetate butyrate (CAB) is used by Ducéré et al. [220]. Like with many other polymers, it has been used in a variety of sensors in conjunction with CNTs and/or other nanoparticles, such as that found in Reference [221], in the work of Yun and Kim. In general, the humidity sensors that rely solely on cellulose compound or its derivatives utilise hydrophilic properties of cellulose, described by Ummartyotin and Manuspiya [222], that result in similar permittivity dependencies as those found in PMMA and PHEMA.

Chemical sensing using polymer technology is a well-established field and further discussion on polymer-based sensing of gasses and other chemicals can be found in the works of Potyrailo and Morris in Reference [223], Korotcenkovi [224] and Potyrailo et al. [225].

### 6.2. Nanomaterial-Based Sensing Materials for RFID Sensors

This section discusses the various nanomaterials that are used in existing RFID sensor implementations. This review will not focus too heavily on nanomaterials that are used for functionalisation or doping but more so on the direct use of specific nanomaterials in RFID sensing. Similarly, this section will not exhaustively describe the synthesis and deposition techniques needed to use these nanomaterials. More information can be found on this topic in the relevant references.

#### 6.2.1. Metal Oxide Semiconductor-Based Sensing

Nanoparticles are particles that have at least 1 dimension less than 1000 nm. Their properties are highly dependent on their size as/and the particle itself is mostly just outer surface [226]. Interestingly, the physical characteristics of specific nanoparticles are not necessarily the same as that found in the bulk material. For example, the hardness of some silicon nanoparticles is 40 times greater than bulk silicon [227]. Also, the characteristics of any specific nanoparticle are dependent heavily on its size, which is defined by the fabrication methodology [226]. The fabrication methods used to manufacture nanoparticles are beyond the scope of this paper and are well documented and discussed by Anu et al. [226], Syafiuddin et al. [228] and in any modern textbook on the subject.

Metal and metal oxide semiconductor (MOS) materials, such as SnO_2_, ZnO and TiO_2_, have been long recognised as ideal materials for sensing gasses, as outlined by Jaaniso and Tan [229], Neri [230] and Bailly et al. [231]. Modern RFID sensors make great use of these materials in nanoscale form both directly, as seen in the works of Sasidharan Nair et al. [75,232] and along with the works of Perret et al. [7] and Vena et al. [233]. These materials have also been used as functionalising materials in the sensor designs of Liang et al. [234] and of Hallil et al. [235]. The use of nano-sized MOS materials over their bulk counterparts is reviewed by Wang et al. [236] and Franke et al. [237], but given the large surface:mass ratio, the sensitivity of the former can be made much greater. The mechanism by which these materials operates as a sensor involves the presence of O_2_ on the surface of the semiconductor, which pulls an electron from the conduction band [226]. A “band bending” effect occurs due to the ions that are now trapped at the surface. With the presence of O (minus) on the surface, gasses such as CO will react and reduce on the surface with this ion and reverse the band bending effect [237]. In polycrystalline semiconductors, this “band bending” effect modifies the Schottky barrier between grains which results in a conductivity change for the solid [237]. The mechanism works for both n-type and p-type MOS materials but with opposing results. This mechanism is also present in other MOS structures, such as nanowires, albeit to a different degree. Examples of this can be seen in the works of Bailly et al. [231], Arafat et al. [238] and Dan et al. [239]. The mechanism by which Si and Si nanowires are humidity-sensitive is reviewed in References [240,241,242,243] and by Chen et al. [244]. It is believed that it behaves similarly to that of MOS materials, in that the ionosorption of the target on the surface results in a change in the charge carrier concentrations [244]. A more in-depth discussion on this content can be found in the works by Wang et al. [236], Jaaniso and Tan [229] and Franke et al. [237]. Not discussed here are the sensing properties of metallic nanowires, but more can be found on this topic in the discussion presented by Chen et al. [244].

A variety of RFID-ready resistive gas sensors have been implemented using MOS nanoparticles, such as TiO_2_, as seen in the works of Pawar et al. [245], Rane et al. [246] and that of Kuang et al. [241]. SnO_2_ nanoparticles have been used for sensing applications by Hallil et al. [235] and Liang et al. [234]. Similarly, SiO2 nanoparticles have been used for sensing purposes by Viespe et al. [247]. Numerous examples exist for Si nanowires, including RFID-based examples, as referenced in the earlier section.

#### 6.2.2. Nanomaterials for Printed Electronics

Printed electronics is one industry that makes extensive use of nanoparticles to fabricate conductive circuits. Some examples of their use can be found in the work of Ghosale et al. [248], Park et al. [249] and Kim et al. [250]. Nanoparticle-based inks have impressive cyclic deformation properties, as discussed by Wang et al. [251], making them suitable for flexible electronics, but care must be taken when selecting the correct nanoparticle to use in an ink. Nanoparticles such as copper nanoparticles are highly susceptible to oxidisation if they are not used with capping agents and the printing process is not carried out in an inert setting [247,252]. For these and other reasons relating to melting temperatures, as outlined by Cui et al. [218], and also cost, silver nano-inks are used above copper or gold nano-inks for printed electronics [248,253]. Alternatives such as graphene nanoparticles can also be used, as presented in Reference [252] by Huang et al. The area of printed electronics is quite vast and is multidisciplinary in nature and cannot be simply summed up in this review. A more in-depth discussion on the topic can be found in the works of Cui et al. [218] and Guo et al. [227].

Such inks have a part to play not only in RFID fabrication but also in RFID sensing, particularly strain sensing. The sensor outlined by Kim et al. [172] was fabricated using silver nanoparticle ink and deposited using a direct stamping transfer method. The use of such a durable fabrication methodology led to this strain sensor supporting strain levels up to 7%, with superior conductive characteristics to that of metallic paste approaches [218]. Further discussion on this deposition methodology for flexible electronics applications can be found in the work of Xiang et al. [254]. The nano-ink used in this paper was comprised of nanoparticles suspended in toluene with a concentration of 5 wt %, similar to that found in the work of Ghosale et al. [248]. It is the use of a stretchable nano-ink that gives this sensor the ability to deform to a much greater degree than is possible with regular copper tracks [172]. Graphene is another emerging nanomaterial that has also gained interest in the area of printable RFID tags. Examples of its use in this area include the work of Kopyt et al. [255], the work of Akbari et al. [256] and that of Scidà et al. [257].

#### 6.2.3. Sensing Properties of Carbon Nanotubes

Carbon nanotubes (CNTs) have been identified as an ideal substance for gas sensing, as reviewed in Reference [258] by Zaporotskova et al. along with References [259,260] by Camilli and Passacantando and Guerin et al. respectively. Temperature sensing prospects have also been thoroughly reviewed by Karimov et al. [261] and also by Saraiya et al. [262]. Similarly, the use of CNTs as coatings for the sensing of pressure/mechanical strain has been well documented by Obitayo and Liu [263]. Several chipless RFID sensors, such as those developed by Baccarelli et al. [264] and Yang et al. [147], leverage the interesting thermal, chemical and electrical characteristics of CNTs. Numerous other sensor types (some wireless) have been implemented using CNT-based sensing films but have not explicitly been defined as RFID sensors, although these implementations could very easily be turned directly into RFID-based solutions. Examples of these implementations include the works of Grow et al. [265], Ong et al. [148], Chopra et al. [266], the work presented in Reference [234] by Liang et al., along with other sensor designs by Penza et al. [136] and by Valentini et al. [267]. A brief review of CNTs is given below, but the sensor designs referenced above utilise either resistive changes or capacitive changes in CNT films when exposed to the appropriate stimulus.

CNTs, like other nanoparticles, are intrinsically hollow, in that they are mostly just surface. Discovered in the 1990s by Sumio Iijima [268], a CNT is a tubular structure that consists of a monoatomic graphene layer as its surface [269]. The cylindrical surface consists of carbon atoms arranged in a hexagonal structure with sp^2^ hybridization [258]. The resulting structure can have the electrical properties of either a semiconductor with a specific bandgap [270] or behave like a metal, depending on the chirality and diameter of the tube [271]. Common growth techniques reviewed by Wang and Yeow [271], result in a mixture of both types of CNTs [272]. CNTs can be found in one of two forms: single-walled (SWCNTs) and multi-walled (MWCNTs). The latter consists of several SWCNTs that surround one another and share the same radial centre. CNTs can be modified through doping or by using functional groups, as discussed by Liang et al. [234], Guerin et al. [260], Ong et al. [148] and in the work of Camilli and Passacantando [259], so as to achieve specific gas/stimulus selectivity. These additions include: carboxyl or amino groups, polymers such as cellulose and PMMA, or other nanoparticles or metal oxides. More information on this topic can be found in the review of CNT sensing properties by Zaporotskova et al. [258]. Although the dielectric properties of CNTs are highly anisotropic, randomly aligned CNTs (deposited) can be used for capacitive style sensing. In this scenario, an applied voltage potential results in strong electric fields build up at the ends of the tubes and the interaction with incoming gas molecules can lead to a polarization of the nearby gas, as described by Wang and Yeow in [271]. Capacitive humidity sensing can be achieved by capillary condensation taking place within the randomly aligned CNT layer [271]. Gas-based conductance changes take place in CNT bundle depositions under a variety of mechanisms. As discussed by Li et al. [273], adsorption of certain molecules leads to a net charge transfer between the molecule (donor or acceptor) and the CNT, which results in an altering of the Fermi level in the band structure, which then results in a conductivity change. This effect is referred to as “intratube modulation” in Reference [273] and gasses such as O_2_ and NO_2_ form this type of contact with the CNT. Other adsorption effects can also take place, such as “intertube modulation”, for gasses such as nitrotoluene (C_7_H_7_NO_2_) [273]. This effect takes place in the interstitial region between SWCNTs and results in the formation of a CNT-molecule-CNT junction, which allows for direct charge transfer between CNTs, whether the CNTs are of metallic or semiconducting type [274]. Other factors of note with regard to CNT-based gas sensing include the need for purging of CNTs when exposed to certain gasses. NH_3_ is one such compound and it is understood that the reason why purging is necessary is caused by the larger degree of chemisorption between NH_3_ molecules and the CNT, as opposed to the more weakly-bound physisorption methods that take place more readily with other molecules. Both chemisorption and physisorption take place between CNTs and a variety of gas molecules but to varying degrees, as described by Calvi et al. [275], Chang et al. in [276] and also observed by Ong et al. in their wireless sensor [148]. The presence of both of these bonding mechanisms can also lead to hysteretic effects, as discussed by Barghi et al. [277]. This topic and its intricacies are beyond the scope of this review but an avid reader can find more relevant information in the works of Zhao et al. [278] and in the discussions found in References [276,277].

Piezoresistive effects present in CNTs, as described by the mechanism(s) presented by Liu et al. [279] and by Anantram et al. in [280], have also been observed in and exploited for CNT film-based sensing of mechanical pressure/strain by Gerlach et al. [281] and by Li et al. [282]. Likewise, piezocapacitive sensors have been fabricated in Reference [283] by Cai et al., where CNT films are placed between parallel plates and deform under stress, in accordance with Possion’s ratio, resulting in a change in capacitance. A similar approach is taken in the sensor implemented by Shin et al. [284], which uses CNT forests instead of CNT films. Both of these approaches rely on the impressive stretchability of CNT networks. A review of the growth mechanisms for these films and some of their resulting properties is presented in Reference [285] by Hu et al. Temperature sensors have also been implemented using MWCNT films that result in thermistor-like behaviour, due to the temperature-dependence of tunnelling mechanisms and phonon-based scattering effects, as described in Reference [286] by Bartolomeo et al.

#### 6.2.4. Sensing Properties of Graphene

CNTs are not the only carbon-based nanomaterial of use in chipless sensing. A complementary split ring resonator (CSRR)-based SIW (Substrate Integrated Waveguide) is coated with a graphene oxide (GO) film in Reference [144] by Chen et al. for humidity sensing purposes. Other papers have also built wireless gas sensors with the help of GO, such as those found in Reference [287] by Le et al., along with the works of Lee et al. [288], Deen et al. [289] and also that of Park et al. [290]. Other sensors based on graphene exist, including wireless bacteria sensors, as outlined by Mannoor et al. [291]. Strain sensors that make use of graphene as a sensing material include the work of Qin et al. [292] and that of Zhao et al. [293], along with the works of Xie et al. [294] and Li et al. [295]. Temperature sensors have also been developed by Davaji et al. [296] and Liu et al. [297] that make use of graphene along with optical and other biological sensors, as reviewed in Reference [298] by Zhao et al. and by Nag et al. [299]. Graphene is effectively considered to be a two-dimensional sheet of pure carbon, arranged in a honeycomb structure, and is equivalent to an unravelled SWCNT and has the same sp_2_ bonding structure [270]. Recently discovered by Andre Geim and Konstantin Novoselov in 2004 [300], this material boasts impressive chemical, physical and electrical characteristics. Similar to CNTs, graphene can be functionalised or doped in a variety of ways to achieve a diverse range of sensing prospects, as described by Nag et al. [299] and Zhao et al. [298], this results in a very diverse range of sensing applications for graphene. The synthesis methods used in the development of graphene are beyond the scope of this review and are discussed in detail in Reference [301] by Choi et al., and also in Reference [299] by Nag et al.

Strain sensing is possible in graphene through several different methods. These include embedding GO in a stretchable polymer matrix, as seen in Reference [295] by Li et al. and also in Reference [292] by Qin et al. Further discussion on the stretchability of the C-C bonds in the embedded graphene can be found in the work of Chen et al. [302]. Other methods involve the use of nanographene films with discrete islands in the film. The tunnelling mechanism between islands is highly dependent on geometry, as discussed in Reference [294] by Xie et al. Further details on graphene-based strain sensing can be found in Reference [303] by Zhao et al.

Graphene and reduced-GO (rGO) have been used for direct humidity sensing by Mannoor et al. [291] and by Shojaee et al. [304]. Other sensors have been outlined that use rGO for the sensing of gasses, such as NO_2_, NH_3_ and CO, such as those found in the works of Le et al. [287] and that of Lu et al. [305]. It should be noted at this point that GO is itself hydrophilic [302] due to the presence of hydrophilic functional groups on its surface [287] and regular graphene exhibits quantum capacitive effects when exposed to humid environments [287,299]. Furthermore, the complete reduction of GO in some cases can be very difficult to achieve [297] and the use of this partial rGO is quite suitable for humidity sensing [304].

Graphene and reduced graphene oxide have also been used for temperature sensing. The former relies on a reduction in electron mobility with increasing temperature [296], although this is dependent on graphene synthesis methods which usually determines whether the resulting graphene will be metallic or semiconducting [297]. In either case, thermo-resistive effects take place but with opposing signs due to a reduction in carrier mobility or due to a thermal activation of charge carriers [297]. Reduced graphene oxide relies on the same mechanisms for thermal sensing.

### 6.3. Other Materials

#### 6.3.1. Cadmium Sulphide (CdS)

A simple light-dependent resistor (LDR) is integrated into a chipless RFID tag by Amin et al. [156]. This type of sensor uses Cadmium Sulphide (CdS), although it behaves like a resistor at low frequencies and like a series RC impedance at microwave frequencies [15,306]. Cadmium Sulphide is a direct bandgap semiconductor material [306] and exhibits photoconductive effects, including within the visible part of the spectrum. The basic principle of operation involves photons being absorbed through the material, that allow some newly energised electrons to jump into the conduction band, thus lowering the effective resistance of the element [306]. Its integration into a quarter wavelength SIR in Reference [156] effectively shorts the open-ended side of the SIR. The result of this addition to the tag results in a change in the resonant characteristics of the SIR when exposed to light. Although this sensing approach was not realised using deposition techniques, such a task would be possible using a CdS film, such as that outlined by Kumar and Meher [306].

#### 6.3.2. Barium Strontium Titanate (BST)

Barium strontium titanate (BST) was used as a temperature-sensitive ceramic material by Mandel et al. [181]. This material has a temperature-dependent ferroelectric polarizability and permittivity arising from a ferroelectric phase transition that takes place at or just below the Curie temperature (depending on the impurity levels in the material), as discussed by Vendik [307]. Impurities can also be added (doping) to allow for alteration of the temperature-dependence of the material [181,308].

#### 6.3.3. Zeolite Materials

Zeolite materials are volcanic minerals that are composed of a tetrahedral arrangement of silicon and aluminium cations along with oxygen anions. Applications of these materials include: catalytic converter materials [309], chemical sensors [310] and wastewater treatment, as discussed by Moshoeshoe et al. [311].

This material was used as a sensitive coating in Reference [138] by Bogner et al. for the sensing of ammonia gas. The complex permittivity of this material is dependent on the presence of ammonia gas, as discussed by Dietrich et al. [312]. The real part is modified by the change in electric polarizability caused by the adsorption of ammonia. The imaginary part sensitivity is dependent on the Si/Al ratio in the zeolite material [312]. Zeolite materials also require purging after exposure to certain gasses, similar to that of CNTs [138].

#### 6.3.4. Absorbant Substrate Materials

Blotting paper was used by Siden et al. [313] as part of their RFID sensor. This material readily absorbs water and this tag design relies on the RF losses present due to the effects of nearby water. Similarly, the tag outlined by Gonçalves et al. [8] uses cork as a dielectric material as its porous structure makes it very sensitive to humidity variations.

## 7. Comparison of Existing RFID Sensors

The RFID sensors presented in the time and spectral domain sections above are presented, along with others in Table 3. New metrics are introduced in Table 3 to evaluate the various sensors that have appeared in the literature. Metrics such as maximum stimulus level and sensitivity were chosen so as to compare but also segregate various sensor implementations from each other. For example, certain strain sensors listed in the table have great sensitivity levels but are not suited to high-strain environmental conditions. Similarly, the metric of measurement time is quite important. As can be seen in Table 3, the use of various coatings for the purposes of temperature and VOC sensing leads to this metric gaining importance as its value significantly differs (where recorded). 

## 8. Discussion

### 8.1. Read Range

Although chipless RFID technology is claimed to be capable of read ranges exceeding one metre, it is not demonstrated to a great degree that many of the presented tags are capable of this feat within the existing spectrum regulations. Several tags have been shown to have the ability to support read ranges of over one metre using UWB IR RADAR technology [153,185,186]. It is unclear whether many of the tags present in this review can in fact be interrogated over such distances without the need for a priori measurements for the given environmental scenario, as described in Reference [27]. With regard to this key metric, more is needed to be done to assess the limitations that certain tag design decisions have on the viable read range of the tag. For example, it is known that heavily meandered antennas do not radiate as effectively as their dipole/monopole counterparts [18,121], likewise the use of dense fractal antennas suffers from similar radiation effects [121,176]. This opens up the possibility that certain spectral domain tags will perform better than others with respect to maximum read range. Such a claim cannot be assessed using the current results presented in the relevant publications as the variation in equipment used and test scenarios cannot be easily accounted for.

### 8.2. Tag Size

Much of the current literature in the area of chipless RFID tags are targeted towards minimizing real estate whilst maximizing address space. This is obviously the case so that the technology may someday challenge the well-known barcode tagging system. As discussed by Machac and Polivka [314], hybrid tags, tags that incorporate multiple resonators, make efficient use of floorspace but parasitic coupling between resonators is a fundamental issue. Some attempts have been made to mitigate this issue, such as the work of Svanda et al. [126], but at significant expense to real estate. Further discussion on this topic is presented in Reference [315] by Svanda et al. From this review, it is clear that the address space of chipless RFID tags is quite large but most of the notable examples do not rely on time domain approaches as they are comparatively quite bulky to spectral approaches, with the exception of SAW tags.

### 8.3. Ease of Fabrication

Arguably the most established design method for chipless RFID sensors is SAW-based tags. Although they boast great address ranges and can be modified for sensor applications, in-situ fabrication of such tags poses a challenge. Many of the other RFID tags are more readily fabricated in-situ using existing printed electronics techniques. However, with regard to sensing, great care is needed when depositing stimulus-sensitive layers to achieve homogeneity and accurate layer thickness.

### 8.4. Tag Location

As described by Dobkin [18] and by many others [152,153], the materials present in the surrounding environment have a significant impact on its performance, particularly in terms of its frequency response. This issue is highly implementation-specific and may render the design of chipless RFID sensors to remain highly application-specific, where no single tag and reader system can suit all environmental applications. Other issues relating to location include orientation issues as many of the current tags rely heavily on linear polarization.

Furthermore, many of the addressing schemes work as expected in single-use settings but it would seem apparent that tags mounted near each other may suffer from signal overlap. This effectively means that the distance between tags and current state-of-the-art beam steering/directing technology impose a limitation on this technology as being suitable for multi-sensor environments.

It also remains to be seen if this type of sensor can indeed be used in extreme environments, given the current materials used to fabricate the tags. This area is of current interest to the authors as many of these sensor types do not require any transistor circuits.

### 8.5. Sensor Performance

Although the performance of various chipless RFID sensors is presented in the above Table (Table 3), several key points of note need to be made:
Many of the humidity and specific VOC sensors exhibit significant cross-sensitivity to undesired stimulii, such as that found in Reference [148]. The majority of other sensors of this type are only tested at a single operating temperature and/or fixed environmental conditions.Some of the materials used in the VOC RFID sensors require purging after exposure to certain stimuli. Examples include those found in References [136,138,148].

## 9. Conclusions

From reviewing the current state-of-the-art in the area of chipless RFID sensors, it is apparent that more work is needed to make this technology into a viable sensing solution. Such work includes further research into reader systems and signal processing methods for regulation-compliant interrogation of various RFID tag designs. Likewise, the metric of read range has not been used to extensively compare various RFID tag design approaches and much of the literature only describes tag performances in laboratory settings. Therefore, further study is needed looking into validating the read range claims that surround the area of chipless RFID. It is the authors’ opinion that this parameter is of critical performance and would need to be done in compliance with existing regulations. This area of research could hopefully identify tag designs that promote read range as opposed to stimulus sensitivity, the more commonly used metric in literature. From the results presented in the above tables, it can be seen that the scattering performance of different spectral domain tags/sensors varies significantly. Although a useful comparison is drawn in the above tables, it is unclear if the variations in test equipment and environmental setup can be neglected fully. Given the fact that the scattering performance of the tag will dominate the sensor read range in real environments, further work is needed to identify spectral tags with good scattering characteristics and to improve those qualities further.

Although the authors would agree with the recent comparison drawn by Herojo et al. [11], the questions of read range and parasitic coupling need to be considered further when utilizing hybrid resonator tags. Work has been done in this area, as discussed above, but that issue is of critical importance because tags of that nature have numerous attractive characteristics, such as compact size and large address space. Similarly, many UWB delay line tags exhibit impressive read ranges, although their designs are comparatively larger than other tag designs. Regardless of this, the literature clearly shows that this approach is quite robust and therefore further efforts should be made to mitigate its low bit-density per unit area.

It would be very interesting to explore the use of chipless RFID sensors in aerospace extreme environments as many of them do not utilize ICs; however, issues relating to nearby conductive materials will need to be overcome to a satisfactory degree. With such an application, it would also be beneficial to push towards a compact, mobile sensor printing system which would deposit the dielectric substrate in-situ on, for example, the element of interest, and then the conductive and stimulus-sensitive elements are deposited over the initial layer, using the same piece of equipment. On the topic of applications and environment, it would be beneficial to implement this technology in a real-life application with a reader and addressing system that is fully capable of operating in a multi-sensor environment. Although this may just reveal pitfalls in the technology, this research into feasibility is currently needed as virtually all of the publications use highly controlled and idealized environments. This may be the case as many of the tags reviewed above are not necessarily designed for remote sensing applications. In any case, the feasibility of this technology for remote sensing needs to be researched further and recommendations need to be made as to how it can be improved, as opposed to just developing new RFID sensing methods.

Many of the cited works hold the opinion that SAW tags are too expensive and too difficult to fabricate in-situ; however, there have been examples of piezoelectric material deposition as referenced above. It would be extremely interesting to investigate if an in-situ fabricated SAW tag has the same impressive characteristic as its current equivalent. This is of great importance as 256-bit addressing has been achieved with this tag design, although it remains to be seen if in-situ fabrication would be an issue for this tag given the tight tolerances needed for the phase encoding scheme used in such tags. Likewise, tags of this type can be readily turned into sensors and several automotive applications already rely on this sensor type.

With regard to the various sensors outlined in this review, many of such designs have issues relating solely to their fabrication materials. These include the need for purging of certain gas sensors, temperature-dependence of many of the permittivity-varying materials and significant stimulus measurement times in some cases. Other sensors that are used for SHM are single-use and very little has been done to suggest that the constituent materials and designs are capable of withstanding effects such as long-term cyclical loading. Although one could assume that this behaviour is guaranteed, it would be extremely worthwhile if such analysis was carried out on the technology, so as to demonstrate the commercial viability of this sensor type.

Although this review attempted to discuss chipless RFID sensor designs and compare in a tabular way, more research, such as that above, is needed to critically compare the various sensor designs more thoroughly.

## Figures and Tables

**Figure 1 sensors-19-04829-f001:**
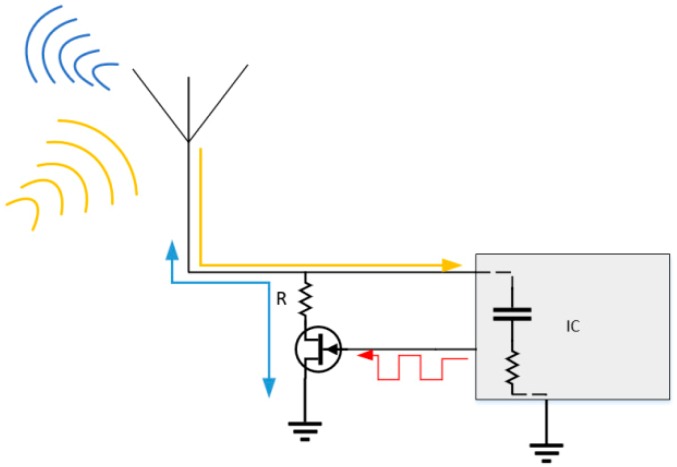
Classic Load Modulation Circuit.

**Figure 2 sensors-19-04829-f002:**
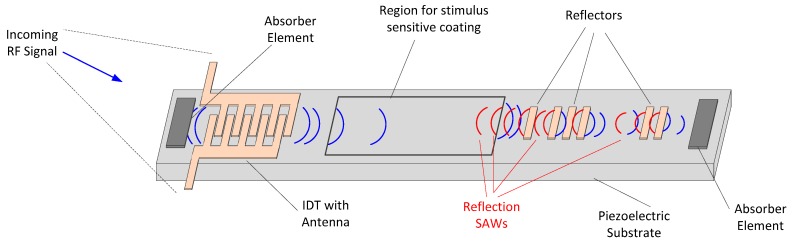
Basic Surface Acoustic Wave (SAW) Sensor.

**Figure 3 sensors-19-04829-f003:**
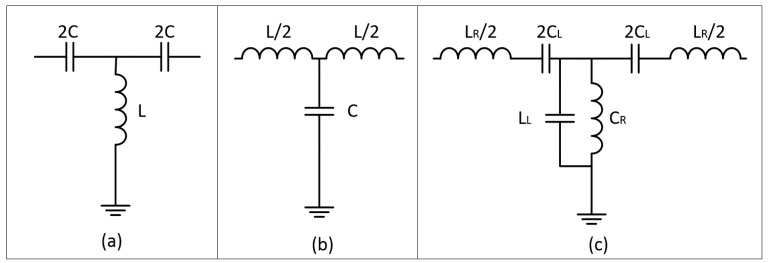
(**a**) Right-Hand (RH), (**b**) Left-Hand (LH) and (**c**) Composite Right-Left Hand (CRLH) Unit Cells—Adapted from Reference [47].

**Figure 4 sensors-19-04829-f004:**
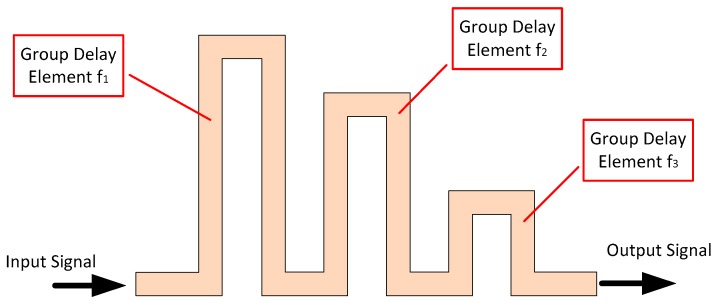
Group Delay Tag Copper Element—Adapted from Reference [52].

**Figure 5 sensors-19-04829-f005:**
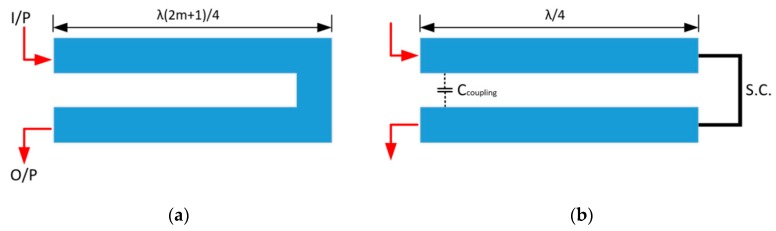
Hairpin Resonator (**a**) and Equivalent Circuit (**b**)—Adapted from Reference [53].

**Figure 6 sensors-19-04829-f006:**
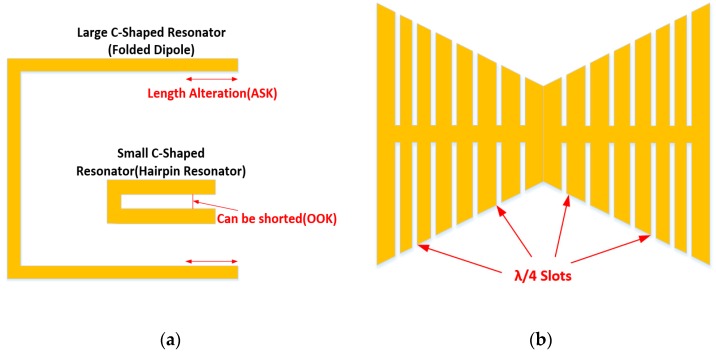
C-Shaped Resonators (**a**)—Adapted from References [93,95]. Slotted Bow Tie Resonator (**b**)—Adapted from Reference [98].

**Figure 7 sensors-19-04829-f007:**
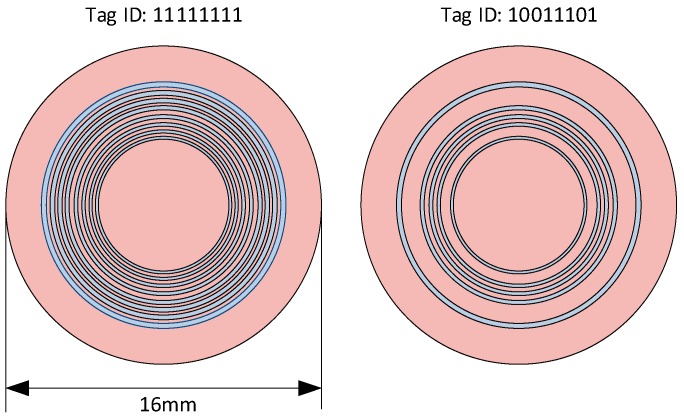
Slot Ring Resonator Tags—Adapted from Reference [106].

**Figure 8 sensors-19-04829-f008:**
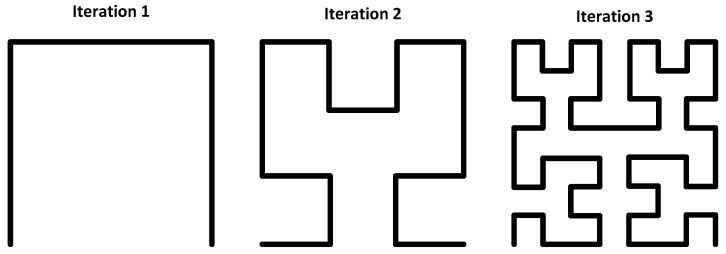
Hilbert Space Filling Curve Geometries (1–3)—Adapted from Reference [123].

**Figure 9 sensors-19-04829-f009:**
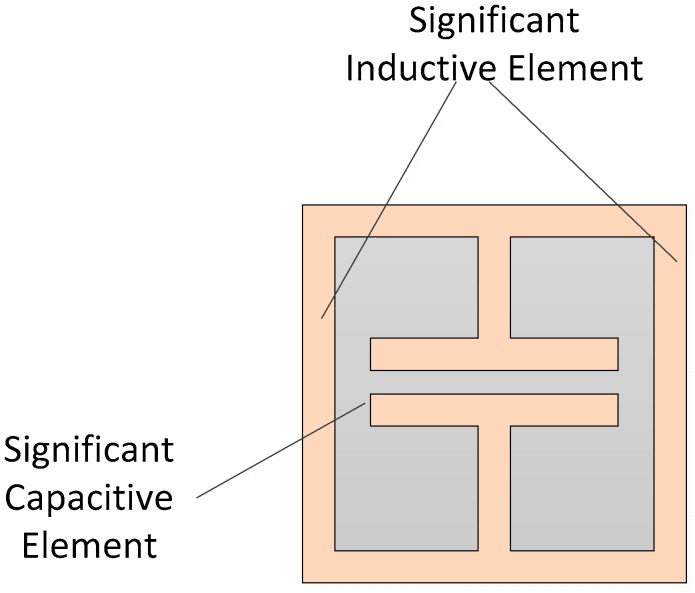
Multi-parameter Sensor Tag—Adapted from Reference [127].

**Figure 10 sensors-19-04829-f010:**
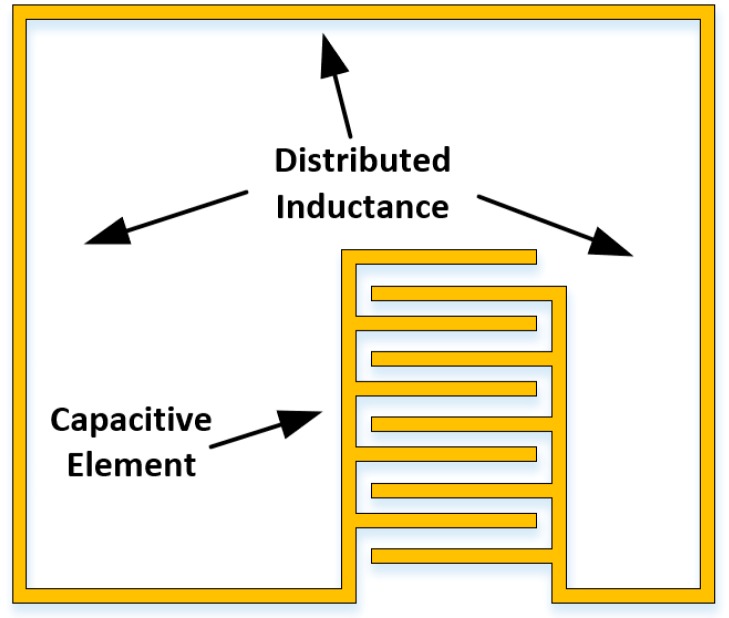
Single Plane Stretchable Resonator Sensor—Adapted from Reference [172].

**Figure 11 sensors-19-04829-f011:**
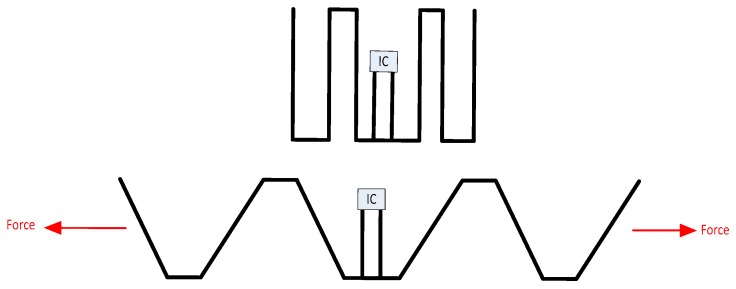
Meander Line Antenna (MLA) Strain Sensor—Adapted from Reference [178].

**Figure 12 sensors-19-04829-f012:**
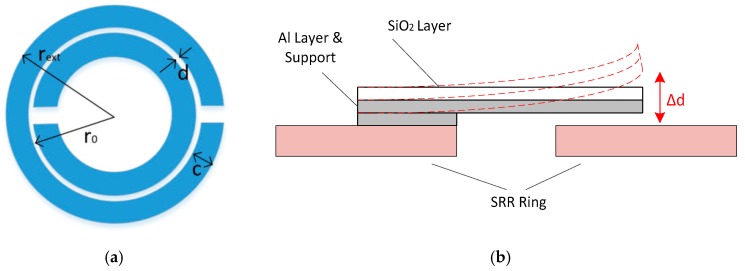
Split Ring Resonator—Adapted from Reference [47] (**a**) and Bi-metallic Strip—Adapted from Reference [183] (**b**).

**Table 1 sensors-19-04829-t001:** Time Domain Tag Comparison.

Tag Design	Year	Major Dimensions	Bits Encoded	Operating Frequency	Interrogation Signal	Reported Read Range	Reader Architecture	Advantages	Disadvantages
SAW	2002 [66]	n/a	64	2.44 GHz	Sinusoidal pulse	n/a	n/a	High Density	Tight manufacturing tolerances
	2004 [67]	n/a	<256	2.45 GHz	Sinusoidal pulse	n/a	n/a	High Density	Tight manufacturing tolerances
	2014 [12]	0.8 × 2.1 mm *	n/a	2–2.5 GHz	LFM Chirp	n/a	n/a	Very Compact	Requires UWB interrogation
	2004 [68]	1 cm length *	n/a	250 ± 50 MHz	Stepped Chirp	n/a	n/a	Robust in multi-sensor environment	Needs accurate reflector design
TFT	2013 [69]	3.9 × 1.5 mm *	16	n/a	Sinusoid	n/a	13.56 MHz RFID Reader	Compact	Uses 222 transistors
	2006 [61]	n/a	64	13.56 MHz	Sinusoid	n/a	13.56 MHz RFID Reader		Uses 1938 TFTs
	2011 [70]	7 × 10 mm	n/a	13.56 MHz	Sinusoid	90 mm	RFID Reader	Works at 13.56 MHz	Uses 1026 TFTs
	2015 [71]	54 cm^2^ *	4	13.56 MHz	Sinusoid	50 mm	RFID Reader	Works at 13.56 MHz	Uses 250 TFTs. Needs large supply voltage
	2017 [57]	6 × 8.3 mm	96	13.56 MHz	Sinusoid	NFC Design	NFC-Enabled Device	Compact	Unclear if it is capable of >1 m read range
	2018 [72]	3 × 9 mm *	n/a	Up to 20 MHz	Sinusoid	n/a	RFID Reader	Suitable for Roll-to-Roll fabrication and 20 MHz operation	Not a completed tag
	2013 [62]	34 mm^2^ *	8	n/a	Sinusoid	n/a	n/a	Compact. Printed with inkjet technology	Not fully integrated into RFID system
Basic Reflection Tag	2008 [41]	Length > 400 mm *	8	1–3 GHz	Sinusoidal pulse	n/a	TDR Module	Simplistic design	Very sensitive to interference
Basic Delay Line	2006 [40]	11.2 × 5.3 cm * (***)	4	915 MHz	Sinusoidal pulse	n/a (no wireless element)	RF Generator	Simplistic design	Bit-density is highly driven by pulse duration. Relatively large. Requires isolators
	2007 [73]	n/a	4	915 MHz	Sinusoidal pulse	25.4 cm	VNA (HP4396B)	Simplistic design	Requires isolators, circulators, etc. Scalability and robustness are questionable
UWB Delay Line	2011 [28]	57 × 70 mm ***	n/a (3 possible values)	3.1–10.6 GHz	UWB Impulse	1.5 m	IR-UWB Radar	No template response needed. Tested at long range	Dual planar, relatively large
	2008 [42]	23 × 31 mm	n/a (3 possible values)	3.1–10.6 GHz	UWB Impulse	0.8 m	n/a	Readily supports IR-UWB Radar interrogation	Dual Planar
CRLH Delay Line	2008 [45]	120 × 14 × 10 mm *	4	Between 2.2 GHz and 4 GHz	Sinusoidal pulse	n/a	n/a	CRLH TL Implemented	Tight tolerances, dispersion issues and high losses occur with CRLH transmission lines
	2009 [44]	Length = 26 cm *	6.2	2.2 GHz	Sinusoidal pulse	n/a	Described in Reference [74]	Implements QPSK	
	2017 [49]	Approximate Length = 12 cm *	2	2.4 GHz	Sinusoidal pulse	14 cm	n/a	More Compact	
Group Delay Tag	2010 [52]	Width > 37.5 mm *	3	2–4 GHz	Chirp or discrete frequencies	n/a	n/a	Fully printable design	Requires UWB interrogation. Odd Harmonics exist, limiting address space
	2013 [75]	10 × 8 cm (2 × 3 cm without antennas)	10 **	n/a	Chirp	30 cm	VNA (PNA N5222A)		
MIW Tag	2012 [76]	Height > 20 mm *	2	2.45 GHz	Sinusoidal pulse	n/a	VNA (E8364B)	Fully Printable	Large losses and tight tolerances for Metamaterial Structures and MIW supporting structures
	2018 [77]	28 × 16 mm *	2	2.9 GHz	Sinusoidal pulse	>1 cm	VNA (N9918A FieldFox)		

* Without Antenna, ** Used as sensor, *** The bit-density of this tag could be much higher if the pulse duration was lower.

**Table 2 sensors-19-04829-t002:** Spectral Radio Frequency Identification (RFID) Tag Comparison.

Tag Design	Year	Major Dimensions	Bits Encoded	Spectral Use	Starting Frequency	$\Delta S_{x1}\;(\mathrm{Max})$	$\Delta S_{x1}\;(\mathrm{Min})$	Read Range	Reader	Polarization	Advantages	Disadvantages
SIR TL	2011 [79]	Length > 100 mm *	2	80 MHz	970 MHz	27	17	n/a	n/a	Linear	Printable, single-plane design	Sub-mm tolerances. Approximately 50 mm per bit encoded
	2016 [84]	5 × 2.5 cm *	1024 (Theory)	5.8 GHz	180 MHz	30	10	2 mm	n/a	n/a	Printable, single-plane design	Requires UWB interrogation
Spiral TL	2009 [88]	88 × 65 mm	35	4 GHz	3 GHz	12	3	5–40 cm	VNA (PNAE8361A)	Cross-Polar	Spirals are compact resonators	Designed for 0.4 m read range. Sub-mm tolerances on spirals
	2010 [5]	n/a	6	1.6 GHz	2 GHz	20	5	10 cm	n/a	Linear	Single Antenna Design. Spectrally efficient	Tight fabrication tolerances on spirals
Spiral TL Group	2018 [126]	14 cm^2^	20	1.2 GHz	2 GHz	15	5	25 cm	VNA (R and S ZVA 40)	Largely omni-directional	Compact, single layer design	66% variation in spectral dips
Stub Loaded TL	2019 [89]	23.8 × 17 mm *	10	1.84 GHz	2 GHz	20	3	n/a	VNA	n/a	Simplistic, compact design	Significant variations in stub insertion loss
	2018 [90]	53 × 34 mm	12	3.25 GHz	3 GHz	35	22	n/a	n/a	Cross-Polarized	Compact and smaller variations in stub insertion loss	Uses a significant amount of the spectrum. Only designed for 0.4 m ranges
	2015 [92]	28 × 20 mm *	18	3 GHz	3.1 GHz	15	12	20 cm	VNA	Cross-Polarized	Very stable stub responses	Uses a significant amount of the spectrum
	2012 [91]	30 × 25 mm	8	2.2 GHz	1.9	25	8	40 cm	VNA (PNA E8362B)	Cross-Polarized	Simplistic Design	Only tested at 0.4 m but response was relatively robust
Resonator Based												
SIR	2014 [82]	42 × 20 mm	8	5.6 GHz including harmonics	3.4 GHz	18	5	20 cm (tests) 40 cm	VNA (PNA E8362B)	Linear	Printable, single-plane design	Uses large amount of spectrum. Only 0.5 m read range recorded
	2016 [83]	55 × 35 mm	46	7.5 GHz	3.1 GHz	8	3	25 cm (tests) 50 cm	VNA (PNA E8362B)	Linear		
Dipole Based	2014 [99]	59 × 17 mm	3	3 GHz	2 GHz	18	7	45 cm (tests) 1 m	VNA (PNA E8362B)	Linear	1 m read range was achieved with 3 dBm Tx power	Relatively large
	2018 [100]	40 × 40 mm	6	4 GHz	3 GHz	7	3	50 cm	VNA	Linear	Simple design with relatively loose tolerances	Not very compact
	2005 [102]	Approximately 50 × 20 mm	5–11	1.1 GHz	5 GHz	5	2	n/a	VNA	Linear	Simple design	Not very compact
	2019 [101]	20 × 20 mm	8	3 GHz	3 GHz	20	10	45 cm	VNA	Omni-directional	Compact. Orientation-independent	Still not as compact as stub loaded TL tags
	2018 [103]	15 × 15 mm	6	3.5 GHz	4 GHz	35 ***	30 ***	n/a	n/a			
	2011 [93]	15 × 15 mm	16	6 GHz	6 GHz	10 *** 4	2 *** 2	n/a	VNA (PNA E8361A)	Dual Polarized	Very compact	Poor spectral response
	2016 [127]	6.8 × 5.5 mm	3	2 GHz	8.5 GHz	10 ***	5 ***	<50 cm	VNA			
	2017 [105]	4.5 × 4.5 mm	4	1.4 GHz	3 GHz	4	2	10–20 cm	USRP	Cross-Polar	Very compact	Poor resonant response
Hairpin/C-shaped	2011 [97]	40 × 20 mm	10	4 GHz	2.5 GHz	18	2	45 cm	VNA (HP 8720D)	Linear	Encodes in phase and frequency	Not very compact
	2016 [96]	26 × 70 mm	20	2 GHz	2 GHz	15	4	30 cm	VNA (ZVA 40)	Linear	Large bit-density	
	2017 [95]	40 × 20 mm	10	3 GHz	2.4 GHz	25	4	n/a	n/a	Linear	More spectrally efficient than earlier tag	
	2018 [94]	121 × 10.5 mm	1	n/a	950 MHz	n/a	n/a	n/a	VNA (PNA-X)	Linear	Operates in ISM band	
Slotted Resonator	2015 [98]	30 × 30 mm	12	7 GHz	3 GHz	4	1	15 cm	VNA (AV 3629D)	Linear	Relatively compact	Poor spectral use/response
	2017 [104]	24.5 × 25.5 mm	36	13 GHz	5 GHz	6	2	n/a	VNA (ZVL13)	Linear	Far more compact	Poor spectral use/response
Ring Resonator	2012 [106]	15 × 15 mm	8	6.5 GHz	6 GHz	10	5	20 cm	VNA (PNA E8361A)	Omni-directional	Very compact	Appears to be parasitic coupling between rings
	2012 [108]	30 × 30 mm	19	7.5 GHz	3.1 GHz	4	2	40 cm	VNA (8722D)	Omni-directional	Very compact	Poor spectral response
	2015 [111]	n/a	8	2 GHz	2.5 GHz	15	5	35 cm	USRP2	Omni-directional	Compact	Apparent low bit-density
	2016 [112]	20 × 20 mm	10	7 GHz	4 GHz	25 ***	15 ***	n/a	n/a	Omni-directional	Compact	
	2018 [109]	<98 × 98 mm	13	5.5 GHz	3 GHz	35 ***	7.5 ***	n/a	n/a	Omni-directional	Robust RCS response	Significant redundancy in design
Grouped Loop Resonators	2016 [115]	20 × 40 mm	28.5	7 GHz	3 GHz	15	5	38 cm	VNA (PNA-LN5232A)	n/a (Bi-directional)	Relatively compact	Only tested up to 30 cm away
Grouped LC Resonators	2011 [120]	150 × 210 mm	10	110 MHz	10 MHz	24	22	21 cm	VNA	n/a	Address can be modified. Low frequency operation	Not very compact
Grouped Rhombic Resonators	2013 [128]	70 × 40 mm	6	3 GHz	3 GHz	11	n/a (ASK)	20 cm	VNA (PNA E8358A)	Linear	Implements ASK—More efficient than OOK	Not very compact
Grouped SRRs	2010 [118]	>18.5 × 8 mm	4	4 GHz	8 GHz	35 ***	25 ***	n/a	n/a	Linear	Simulation and Testbed response appear good	Sub-mm fabrication tolerances (0.0 × mm). Some coupling still exists
Space Filling Curves	2006 [129] 2010 [123]	Approximately 150 × 30 mm	5	1.5 GHz	3 GHz	5	2	1.22 m	VNA (E-5071B)	n/a—supports bi-directional	Good use of spectrum space	Poor measured resonant response
SIW Resonator	2019 [130]	Approximately 25 × 20 mm	n/a	13 GHz	22 GHz	35	15	10 cm	n/a	n/a orthogonal	Less regulations at these frequencies	Microstrip elements have high losses at these frequencies. More expensive reader required

* Without antennas attached, *** In simulation only. The values in a real implementation can be significantly lower.

**Table 3 sensors-19-04829-t003:** RFID Sensor Comparison Table.

Sensor Stimulus	Year	Tag Type	Major Dimensions	Max Stimulus (Range)	Sensitivity	Comments	Measurement Time
Strain	2014 [172]	Resonant	30 × 30 mm	7%	15–52 MHz/%	Non-linear sensitivity up to 7% but smaller linear regions exist	n/a (Structural or SAW)
	2011 [175]	Antenna Impedance	36 × 36 mm (No strain)	6%	0.429 dB/%	Read range dependent on strain. Includes IC	
	2012 [9]	Resonant	100 × 100 mm (Approximate)	0.2%	915 MHz/%		
	2017 [168]	Resonant	6.3 × 6.3 mm	21,300 µε	105.6 MHz/ε		
	2009 [170]	Resonant	15 mm^2^	6000 µε	5.148 MHz/ε		
	2015 [10]	SAW/Resonant	10 × 3 mm *	130 µε (test limit)	−1.8 ppm/µε	Includes addressing scheme. Highly temperature sensitive	
	2019 [178]	MLA/Resonant	35 × 15 mm (Approximate)	50%	1.2 MHz/%	Tested up to 7.5 m range. Read range dependent on strain. Includes IC	
Crack Detection	2018 [167]	Resonant/Impedance	62 × 62 mm	5 mm	−3.4 MHz/mm		
	2018 [166]	Resonant	n/a (Space Filling Curve)	Single fracture	4 MHz/crack		
Temperature	2009 [179]	SAW velocity	n/a	190 °C	n/a (see Reference [164])		n/a
	1998 [159]	SAW velocity	n/a	85 °C	6.03 kHz/°C		n/a
	2011 [181]	Transmission Line Termination	n/a	100 °C	0.5°/°C	Phase and magnitude encoding	n/a
	2012 [233]	Resonant/Si NWs	12 × 15 mm (Approximate)	19 °C	1.625 MHz/°C (With humidity Change)		120–255 seconds
	2018 [180]	Resonant/Substrate-based	46 × 20 mm (Approximate)	100 °C	n/a		n/a
	2015 [139]	Resonant/Graphene Oxide	n/a	40 °C	−7.69 kHz/°C		n/a
	2016 [127]	Resonant/Phenanthrene	6 × 6 mm	72 °C	Threshold-based—320 MHz change	Single Use	60 min
	2019 [184]	Resonant and Q factor	25 × 25 mm	4, 8, 12, 16 °C Thresholds	Threshold-based		n/a
	2010 [183]	Resonant/Bi-metallic Strip	3 × 3 mm * (**) (Approximate)	Up to 300 °C	Up to 780 MHz/°C	Operates between 28 and 34 GHz. Coupled to CPW which has not been included in dimensions	n/a
	2012 [185]	UWB Delay Line/Termination	10 × 13.65 cm	25 °C–130 °C	(Approximately) −0.191 dB/°C	Uses SMD termination load. Good read range	0–3 min
	2015 [186]	UWB Delay Line/Termination	83.5 × 78.4 mm	n/a	50 °C Threshold	Uses external thermostat sensor	n/a
Pressure	2013 [134]	SAW—Resonant	5 × 5 x 5 mm *	4 Bar	0.33 ± 0.02 MHz/Bar	Some hysteretic effects take place. Operates between 10 and 14 GHz	n/a
	2006 [163]	SAW—Resonant	Tire valve size (assumed)	150 Psi	Sub 0.4 Psi resolution	n/a	n/a
	1998 [159]	Saw-Resonant/Velocity	15 × 15 mm *	10 Bar	8.33 kHz/Bar (Estimated)	n/a	n/a
	2013 [164]	OFET/(PMOFET)	45 × 20 mm **	78.125 Mpa	30.7 mA/Mpa	n/a	n/a
	2009 [169]	Resonant/Diaphragm	5.8 × 3.8 × 1.4 *	3 Bar	370 MHz/Bar	n/a	n/a
Humidity	2018 [103]	Resonant/PVA	15 × 15 mm	n/a	n/a	Operates between 4.5 and 7.5 GHz	
	2006 [135]	SAW—Resonant and insertion loss/PVA	n/a	30–90% range	0.1625 dB/%	Some hysteresis occurs with PVA. Resonant changes were not very linear	n/a
					1.78 kHz/%		
					3 kHz/%		
	2013 [75]	Group delay and resonant/(Si NWs)	98 × 63 mm	60.2–88% range	0.802 ns/%	All variations were non-linear	
	2016 [127]	Resonant/PVA	6 × 6 mm	35–85%	5.714 MHz/%		n/a
	2018 [144]	Resonant/Graphene Oxide	8.5 × 6 mm	11.3–97.3%	772.6 kHz/% below 84.3%	Low hysteresis levels	n/a
					3.415 MHz/% above 84.3%		
	2015 [139]	Resonant/Graphene Oxide	10 × 10 mm	55–95%	17.8 kHz/%		n/a
	2018 [100]	Resonant/Si NWs	40 × 40 mm	30–90%	2.9 MHz/% below 70%		10 min
					4.8 MHz/% above 70%		
	1997 [132]	FET/Polyanailine	Single FET	15–50%	0.9615 µA/s/%	Turn on current sensitivity is the variable	n/a
	2017 [145]	(TL) Resonant spiral/PVA	>20 × 10 mm *	21–53%	2.5 MHz/%		n/a
VOC Gasses	2004 [136]	SAW-Resonant/CNTs	3 × 1 mm *	Ethanol: 1.3–180 ppm	Ethanol: 6.89 kHz/ppm	Serious recovery issues once stimulus removed	2–4 min
				Toulene: 2.8–180 ppm	Toulene: 7.47 kHz/ppm		
				Ethylacetate: 2.7–180 ppm	Ethylacetate: 5.45 kHz/ppm		
	2002 [148]	Resonant/CNTs	20 × 20 mm	CO_2_: 20–80%	CO_2_: %(Δε) = 1/60%	Humidity and temperature variations have unwanted effects. Irreversible NH_3_ effects	CO_2_: 45 seconds
				O_2_: 0–100%	O_2_: %(Δε) = 0.07/100%		
							O_2_: 4 min
				NH_3_: 0–100%	NH_3_: n/a		NH_3_:2 min
	2009 [147]	Resonant-Reflection/CNTs	118 × 27 mm	n/a	n/a	10.8 dBi effect of NH_3_ presence	
	2016 [146]	Resonant/Porous Substrate	8 × 10 mm *	Acetone: 0.1%	Acetone: 0.875 GHz/%	Desorption time > 15 min	n/a
				Methanol: 0.04%	Methanol: 1.5 GHz/%		
	2017 [138]	Resonant/Zeolites	>4 × 4 mm *	n/a	<0.5 GHz for various stimuli	Significant purging required after use. Regeneration of zeolites required	10–20 min
	2013 [137]	Resonant-Impedance/PEUT	n/a	0–27% RH	5.926 kHz/%	Allows for temperature compensation	n/a
	2008 [133]	TFT-(Drain-source current)/rr-P3HT	n/a	n/a	%ΔI_DS_ =		1–2 min
					Acetone: 0.57%/100 ppm		
					Toluene: 0.2%/280 ppm		
					Butanol: 0.6%/75 ppm		
Orientation/Rotation	2019 [158]	Polarization	n/a	360°	Change of 40 dBm in RCS	0–180 results are equal to 180–360 results. Not linearly sensitive	n/a
	2017 [157]	Resonant/Cross-Polar response		95°	Non-linear magnitude and frequency change		n/a
Permittivity/Velocity	2007 [50]	CRLH resonant and phase	n/a	ε = 1.027 led to 180 MHz shift (2500°/ε)	n/a		
	2009 [51]	CRLH Capacitive	n/a	60° shift experienced for placed item on belt	Phase-based measurement	Measurement issues exist	n/a
	2016 [171]	Resonant-Read Range/substrate permittivity	39.5 × 25 mm	ε = 2.5–8	n/a	Non-linear RCS response. Operates at a single frequency	n/a
	2018 [152]	UWB Delay Line	30 × 30 mm (per cell)	(Non-linear) 20 GHz/%$\varepsilon_{r}$	n/a (Tested up to $\varepsilon_{r}$ = 5)	Relies on pre-set polarization	n/a
	2018 [153]	Resonant	>20 × 22 mm (per cell)	−8.474 GHz/%$\varepsilon_{r}$	n/a (Tested up to $\varepsilon_{r}$ = 4.54)	Read range enhanced with redundant elements	n/a
	2012 [154]	Resonant Stub	14 × 14 cm	−2.375 GHz/%$\varepsilon_{r}$	n/a (Tested up to $\varepsilon_{r}$ = 4)	Unclear if sensor response is linear	n/a
	2015 [155]	UWB Delay Line—Mode Delay	58 × 102.5 mm	−16.67 ns/%$\varepsilon_{r}$	n/a (Tested up to $\varepsilon_{r}$ = 4.24)		n/a (Negligible)

** This was only the setup size—it could be made more compact, * Without Antenna.

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
