# Peer review of "A Review of Chipless Remote Sensing Solutions Based on RFID Technology"

_sensors, 2019, doi:10.3390/s19224829_

Round 1

Reviewer 1 Report

This paper presents  Chipless Radio Frequency Identification (RFID), used in a variety of remote sensing applications. There is no clarification on difference between  chipless and chipped RFID sensor.  A notation list is necessary.  Apart from just ID recognition, the team had postulated uses for environmental sensors. But is is not clear whether all without using chip technology.  

Regarding the “Material” theme seems very superficial specially identification of suitable materials, or “investigation of new materials”. It is difficult to assess the novelty.  Experimental results should be presented in the paper eg. various  sensing materials eg. nano-materials, graphene incorporated in RFID sensor should be discussed. 

While the simple conclusion is not adequate from an immediate technological perspective, a more detailed discussion/conclusion is in order particularly in the context of any other relevant reported studies of similar or dissimilar materials. Sufficient information is not included or cited to support assertions made and conclusions drawn.  Discussion and conclusion need to be enhanced satisfactory.

Reviewer 2 Report

I think it's an excellent and extensive review of Chipless RFID technology.

However, I have some comments.

In my opinion, UWB time-domain tags and sensors are not treated with enough detail. For example, in section 3.2 (Delay line tags) only is treated tags based on metamaterials.

For example, in section 3.2 only few references about tags based on UWB delay lines are included. Although this chipless technology has some limitations in the number of bits, it enables to reach higher distance than frequency-coded tags and it can be used with standard UWB radars. In table 1 a row about UWB time-domain tags will be included. I suggest to include some works such as:

“A balloon-shaped monopole antenna for passive UWB-RFID tag applications,” IEEE

Antennas and Wireless Propagation Letters, Vol. 7, 366–368, 2008

"Chipless UWB RFID tag detection using continuous wavelet transform." IEEE Antennas and Wireless Propagation Letters 10 (2011): 520-523.

"Time-domain measurement of time-coded UWB chipless RFID tags." Progress In Electromagnetics Research 116 (2011): 313-331.

In these works, the theory of time-domain tags and a method for detection of chipless time-domain tags is presented allowing the detection of tags to 2-3m using commercial UWB radars.

In addition, some temperature passive sensors based on time-domain have been reported in the literature:

"Passive wireless temperature sensor based on time-coded UWB chipless RFID tags." IEEE Transactions on Microwave Theory and Techniques 60.11 (2012): 3623-3632.

"Signal processing techniques for chipless UWB RFID thermal threshold detector detection." IEEE Antennas and Wireless Propagation Letters 15 (2015): 618-621.

Some recent works about permittivity sensors for structural health based on time-domain and frequency-coded chipless RFID are not included:

 "A depolarizing chipless RF label for dielectric permittivity sensing." IEEE Microwave and Wireless Components Letters 28.5 (2018): 371-373.

"Chipless dielectric constant sensor for structural health testing." IEEE Sensors Journal 18.13 (2018): 5576-5585.

"Passive wireless permittivity sensor based on frequency-coded chipless RFID tags." 2012 IEEE/MTT-S International Microwave Symposium Digest. IEEE, 2012.

"Wireless concrete mixture composition sensor based on time-coded UWB RFID." IEEE Microwave and Wireless Components Letters 25.10 (2015): 681-683.

Readers. In my opinion, the authors should include a section about readers used and calibration/measurement techniques. Most of papers based on frequency-code and resonators used expensive VNA for the measurements. In some cases, background subtraction is required, therefore these tags are not practical because require some human action or only can be read at very short range with expensive antennas that probably are in near field region. Examples of commercial time-domain UWB radar are given in the suggested references. The cost of the reader is one of the most important drawbacks of chipless technology over chip-based RFID. Some custom readers have been proposed in the literature for chipless frequency-coded tags and time-domain. Some comment should be included about the readers in the text.

Some comments about mmwave chipless tags (eg. based on SIW resonators at 24 GHz) can be included.

I suggest to include the read range and reader used in tables 1-2. These parameters are important. The authors enter in too much detail in Appendix A about one special chipless type. As the paper is too long, I suggest to remove or summarize the appendix A.

Minor mistakes:

In line 187, Appendix 1 must be Appendix A

Round 2

Reviewer 2 Report

The authors have been taken into account my comments. Therefore, I have no additional comments.